



# Updating the radiation infrastructure in MESSy (based on MESSy version 2.55)

Matthias Nützel[1,a], Laura Stecher[1], Patrick Jöckel[1], Franziska Winterstein[1], Martin Dameris[1], Michael Ponater[1], Phoebe Graf[b], and Markus Kunze[2]

[1]Deutsches Zentrum für Luft- und Raumfahrt, Institut für Physik der Atmosphäre, Oberpfaffenhofen, Germany

[2]Leibniz Institute of Atmospheric Physics (IAP), Kühlungsborn, Germany

[a]now at: Meteorologisches Institut München, Ludwig-Maximilians-Universität München, Munich, Germany

[b]previously at: [1]

**Correspondence:** Matthias Nützel (matthias.nuetzel@dlr.de)

**Abstract.**

The calculation of the radiative transfer is a key component of global circulation models. In this manuscript we describe the most recent updates of the radiation infrastructure in the Modular Earth Submodel System (MESSy). These updates include the implementation of the PSrad radiation scheme within the RAD submodel. Further, the radiation-related submodels

5   CLOUDOPT (for the calculation of cloud optical properties) and AEROPT (for the calculation of aerosol optical properties) have been updated and are now more flexible in order to deal with different sets of shortwave and longwave bands of radiation schemes. In the wake of these updates a new submodel (ALBEDO), which features solar zenith angle dependent albedos and a new satellite-based background (white-sky) albedo, was created. All of these developments are backward compatible and previous features of the MESSy radiation infrastructure remain available. Moreover, these developments mark an important

10  step in the use of the ECHAM/MESSy Atmospheric Chemistry (EMAC) model as the update of the radiation scheme was a key aspect in the development of sixth generation of the the European Centre for Medium-Range Weather Forecasts – HAMburg (ECHAM6) model from ECHAM5 and they also aim towards the use of MESSy with the ICOsahedral Non-hydrostatic (ICON) model. The improved infrastructure will also aid in the implementation of additional radiation schemes once this should be needed.

We have optimized the set of free parameters for two dynamical model setups for pre-industrial and present-day conditions: one with the radiation scheme that was used up to date (i.e. the radiation scheme of ECHAM5) and one with the newly implemented PSrad radiation scheme. After this parameter optimization, we performed four model simulations and evaluated the corresponding model results using reanalysis and observational data. The most apparent improvements related to the updated

20  radiation scheme are the reduced cold biases in the tropical upper troposphere and lower stratosphere and the extratropical lower stratosphere, and a strengthened polar vortex. The former is also related to improved stratospheric humidity and its variability if the new radiation scheme is employed.





Using the multiple radiation call capability of MESSy, we have applied the two model configurations to calculate instan-
taneous and stratospheric adjusted radiative forcings related to changes in greenhouse gases. Overall, we find that for many
forcing experiments the simulations with the new radiation scheme show improved radiative forcing values. This is in particular
the case for methane radiative forcings, which are considerably higher when asessed with the new radiation scheme and thus
in better agreement with reference values.

# 1 Introduction

The most accurate models for calculating the radiative transfer within the atmosphere are line-by-line (LBL) models (e.g. Pin-
cus et al., 2015). Results from radiative transfer calculations with these models agree well with observations (e.g. Oreopoulos
and Mlawer, 2010; Oreopoulos et al., 2012; Pincus et al., 2015, and references therein). The shortwave (SW) and longwave
(LW) broad band errors of LBL models are in the order of $1\,\mathrm{W\,m^{-2}}$ (Pincus et al., 2015, and references therein). However,
these detailed radiative transfer models are computationally too expensive to be run in global climate models (e.g. Oreopoulos
et al., 2012). Hence, in global climate models the radiative transfer calculation is simplified compared to LBL models (e.g.
Oreopoulos et al., 2012) and it is also, typically, not performed every time step (Pincus and Stevens, 2013). In total, there is the
challenge for these simplified radiative transfer codes to be sufficiently precise and efficient (Pincus and Stevens, 2013). This
causes the need to revise the radiation schemes which are employed in global models from time to time.

Here, we describe how we extended the Modular Earth Submodel System (MESSy; Jöckel et al., 2005, 2010) infrastruc-
ture to include the PSrad radiation scheme (Pincus and Stevens, 2013) for further use in MESSy-based climate simulations.
The previous status of the MESSy radiation infrastructure is evident from Dietmüller et al. (2016). They document how the
radiation infrastructure of the fifth generation European Centre for Medium-Range Weather Forecasts – HAMburg (Roeckner
et al., 2003, 2006, ECHAM5;) model was restructured to be "MESSy-fied", i.e. to be modularized according to the MESSy
coding standard: new (MESSy) submodels have been created from code parts of the radiation calculation which are related
to, but to a certain degree independent of, the radiation scheme. These new submodels were (i) AEROPT for the provision of
aerosol optical properties, (ii) CLOUDOPT for the calculation of cloud optical properties and (iii) ORBIT to determine the
orbital parameters, which are needed e.g. for the calculation of the radiative transfer. During this process also the structure of
the radiation scheme was "MESSy-fied" and the corresponding MESSy submodel RAD was created.

Besides the pure modularization, Dietmüller et al. (2016) also describe that the MESSy radiation infrastructure provides
additional valuable features connected to the radiation calculation. One example is the possibility for multiple (diagnostic)
calls of AEROPT, CLOUDOPT and RAD, which can be used to determine multiple instantaneous radiative forcings (RFs) or



stratospheric adjusted radiative RFs (as described by Stuber et al., 2001) online in a single simulation (see e.g. Hansen et al., 2005, for a definition of instantaneous and adjusted RFs). This is a powerful feature as the need for extensive output, which would be required for an offline (post-simulation) calculation, is avoided and (if intended) all calculations are consistently performed with exactly the same version of the radiation scheme. Further, the diagnostic calls are performed under the same meteorological conditions and at the highest possible frequency, i.e. the frequency of the radiation calls. Hence, a major con-

cern during the development phase, which is described here, was to secure backward compatibility (up to the degree of binary identity to some point) and the possibility of retaining these features in connection with the newly added radiation scheme. Besides the integration of an additional radiation scheme, we also made the radiation infrastructure more flexible. Moreover, we created the MESSy submodel ALBEDO, which now contains the previous code for the calculation of the surface albedo extracted from the RAD submodel and newly added parametrizations for the calculation of the surface albedo.


As mentioned above, until now the default radiation scheme in MESSy was a modularized version of the ECHAM5 radiation scheme, which we will denote by E5rad throughout this paper. For many years the Max Planck Institute for Meteorology (MPI-M) in Hamburg, Germany, has developed the general circulation model ECHAM (e.g. Roeckner et al., 1996, 2003; Stevens et al., 2013). An important step from the fifth generation of ECHAM to ECHAM6.1 was an update concerning

the radiation scheme, in particular as in the SW the number of bands was increased from 4 to 14 (Stevens et al., 2013). For the latest version of ECHAM, ECHAM6.3, the LW and SW radiation parametrization was once more revised as the PSrad scheme (Pincus and Stevens, 2013) was made available (Giorgetta et al., 2018; Mauritsen et al., 2019). This version of ECHAM - with PSrad as the radiation model - also constitutes the atmospheric component of MPI-ESM1.2, the MPI-M's Earth System Model, which is described by Mauritsen et al. (2019). Simulations with MPI-ESM1.2 have con-

tributed to the most recent phase of the coupled model intercomparison project (CMIP6; Eyring et al., 2016, see https://pcmdi.llnl.gov/CMIP6/ArchiveStatistics/esgf_data_holdings/, accessed last 10 Jul 2023, for a list of available model output).

Similarly, PSrad is the radiation scheme employed in an ICOsahedral Non-hydrostatic (ICON originally developed by the

Deutscher Wetterdienst, DWD, and the MPI-M Zängl et al., 2015) model version described by Giorgetta et al. (2018). We decided to add the PSrad scheme (as implemented in ICON version 2.4.0) to the radiation schemes, which are available within the MESSy infrastructure. This update marks an important step to incorporate previous model developments of the ECHAM family within the MESSy infrastructure, while it is also an important step towards the use of ICON as a base model within the MESSy infrastructure.


Further, we expect a reduction or removal of previous shortcomings related to the old radiation when employing the new radiation scheme PSrad in ECHAM/MESSy Atmospheric Chemistry (EMAC) simulations. For example, previous studies with EMAC and the ECHAM5 radiation scheme have shown considerably low radiative effects for methane. For instance, a doubling of the present-day reference value of $1.8\,\mu\mathrm{mol\,mol}^{-1}$ resulted in $0.23\,\mathrm{W\,m}^{-2}$ (Winterstein et al., 2019; Stecher et al.,



2021), while studies of Myhre et al. (1998) and Etminan et al. (2016) suggest $0.53\,\mathrm{W\,m^{-2}}$ and $0.62\,\mathrm{W\,m^{-2}}$, respectively, for doubling of the reference value of $1.7\,\mu\mathrm{mol\,mol^{-1}}$.

In the following, we present the recent developments concerning the radiation-related MESSy submodels, AEROPT, CLOUDOPT, RAD and the new MESSy submodel ALBEDO (Sect. 2). The dynamic atmosphere-only model setups (i.e. no interactive

aerosol, only simplified methane chemistry) driven either by the new (PSrad) or the old (ECHAM5) radiation scheme, as well as the parameter optimization process are presented in Sect. 3. This section also features the evaluation of these model setups with observational and reanalysis data. In Sect. 4 we show RF estimates derived using the old and new radiation scheme and we compare our results to results from previous studies (Sect. 4). Finally, we close with a summary of the presented results (Sect. 5).

## 100  2  Radiation infrastructure updates

### 2.1  MESSy (short description)

Here, we describe the updates of the radiation infrastructure of the Modular Earth Submodel System (MESSy; Jöckel et al., 2005, 2010), which are now implemented in MESSy based on version 2.55. MESSy is a middleware to link different submodels (e.g. representing physical processes, chemical processes, online diagnostics, or external couplers) with a base (dynamical

core) model. The key concept behind MESSy is that it provides the general infrastructure to perform simulations with a specific base model and clean interfaces, which allow the coupling of different submodels to this base model (Jöckel et al., 2005) or even the internal coupling of different modelling compartments (Pozzer et al., 2011). The software layers to ensure this clear separation are the base model layer (BML) and base model interface layer (BMIL), which contain the base model's code and the interface to connect submodels to the base model, respectively (Jöckel et al., 2005). Similarly, submodels are split into two

layers, which contain the core of the submodels computations in the submodel core layer (SMCL) and the submodel interface layer (SMIL) to connect with other submodels or the BMIL (Jöckel et al., 2005). The exchange of variables (between submodels etc.) is handled via the "CHANNEL" interface (Jöckel et al., 2010) to avoid compile time dependencies between the core routines of different submodels.

Dietmüller et al. (2016) describe the state of the MESSy radiation infrastructure at the starting point of our new implementations. With the radiation infrastructure update and development of the submodels AEROPT, CLOUDOPT, RAD and ORBIT, they made a big step towards a clean separation between (i) code components that are relevant for the radiation calculation but which can be separated, from a software development perspective, from the core radiative transfer model and (ii) the core radiative transfer model. Of course, it must be still ensured that the input and output variables of the submodels connect prop-

erly: for example, if the SW scheme has a set of bands, AEROPT and CLOUDOPT must provide aerosol and cloud optical properties for exactly those bands. Consequently, with the introduction of an additional radiation scheme, we had to update the submodels AEROPT and CLOUDOPT as the band structure of the newly added radiation scheme, PSrad, differs from the old




one (see Sect. 2.2). Further, we conducted an additional separation of code that is independent of the core radiation calculation by creating the new MESSy submodel ALBEDO for the calculation of the surface albedo, which is then provided as an input

for the radiation scheme. A key requirement of the updates was to preserve the previous flexibility, in particular concerning the application of multiple calls of the radiation scheme as well as multiple calls of the aerosol and cloud optical schemes, as described by Dietmüller et al. (2016). Figure 1 gives an overview of the new radiation infrastructure. The following sections describe the updates for each of the radiation infrastructure submodels (i.e. all submodels that are directly related to calling the radiation scheme) in MESSy in comparison to the state described by Dietmüller et al. (2016). The described changes are

available at the latest in ALBEDO version 1.4, AEROPT version 2.1.0, CLOUDOPT version 2.5 and RAD 3.0.

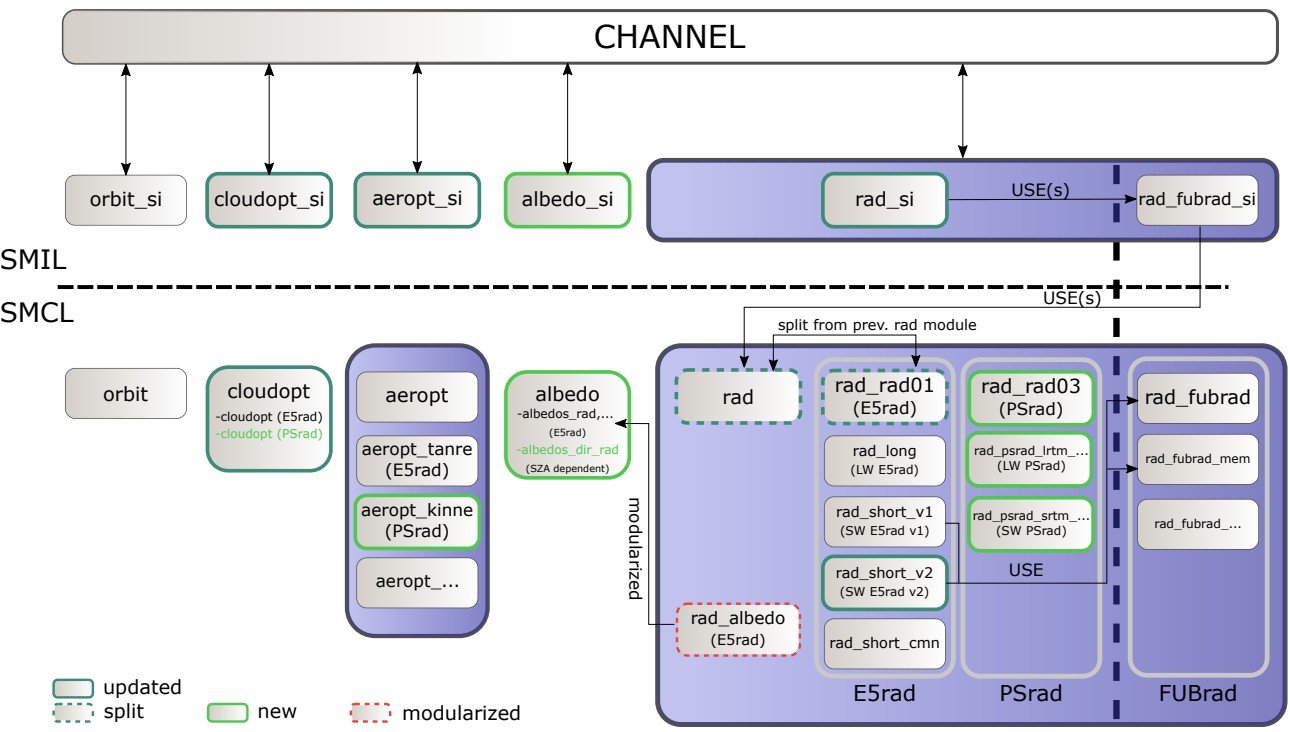

**Figure 1.** Schematic overview of the updated MESSy radiation infrastructure in comparison to the state described by Dietmüller et al. (2016; see also their Fig. 1). Green colour indicates new submodels (either Fortran modules or Fortran subroutines). Individual Fortran modules are shown as grey boxes. If the MESSy submodels encompass more than one Fortran module this is indicated via blueish boxes. See text for details. In addition to the depicted changes, additional minor modifications, e.g. in the AEROPT core layer modules, have been made during the revision of the radiation infrastructure.



## 2.2 RAD: updates of the MESSy radiation submodel

The submodel RAD calculates the radiative transfer taking into account aerosols, clouds, and selected gaseous species relevant for radiative transfer (Dietmüller et al., 2016). Based on Dietmüller et al. (2016), we give the following recap of the RAD
submodel before our implementation: In RAD a MESSy-fied version of the ECHAM5 radiation scheme is available. This module comprises a LW radiation scheme with 16 bands (RRTM; Mlawer et al., 1997) and two SW schemes (short_v1 and short_v2) with four bands each, both based on Fouquart and Bonnel (1980), whereas short_v2 includes the improvements of Thomas (2008). The MESSy submodel FUBrad (Nissen et al., 2007; Kunze et al., 2014) can be switched on to overcome the relatively coarse resolution in the SW, which allows for high-resolution UV radiative transfer calculations in the stratosphere
above 70 hPa, and extends the spectral range by including $O_2$ UV absorption in the Schumann-Runge bands/continuum and Lyman-$\alpha$.

Here, we implemented the radiation scheme PSrad (Pincus and Stevens, 2013), as available in ICON version 2.4.0, into the MESSy submodel RAD. As described by Pincus and Stevens (2013), the development of PSrad was guided by RRTMG
(Mlawer et al., 1997; Iacono et al., 2008). RRTMG in turn features 16 bands in the LW and 14 bands in the SW (Iacono et al., 2008; see also Tables 2.3 and 2.4 presented by Giorgetta et al., 2013b, for the band structure). To make the PSrad scheme available alongside the "old" schemes, we introduce a new software "layer" in the MESSy RAD submodel core by splitting the previous core module "messy_rad.f90" into two new Fortran modules "messy_rad.f90" and "messy_rad_rad01.f90", where the latter contains all subroutines from the previous "messy_rad.f90" directly related to the ECHAM5 radiation scheme(s).
In analogy to "messy_rad_rad01.f90" for the old radiation scheme, "messy_rad_rad03.f90" provides the interface to the new radiation scheme (PSrad). The Fortran module was numbered with "rad03" as in the SW rad01 already contains 2 schemes rad_short_v1 and rad_short_v2. In principle, the two LW and three SW schemes can be combined freely and the introduction of additional schemes should be straightforward, if they are well modularized. However, for new combinations additional parameter optimization (see Sect. 3.2) will likely be required. While it is still possible to use FUBrad with the old SW radiation
schemes, this is not yet possible with the new SW scheme. In a future step, also the new SW scheme is envisaged to be available in combination with the FUBrad submodel. At the moment, however, the model terminates with a controlled shutdown and a corresponding error message, if this combination is selected.

This implementation marks a major update of EMAC as one key update between ECHAM5 and ECHAM6 was the update
of the (SW) radiation scheme (Stevens et al., 2013), which in ECHAM6.3 was updated to PSrad (Giorgetta et al., 2018; Mauritsen et al., 2019). Further, it also marks an important step for the transition towards ICON as a MESSy base model as we implemented PSrad as available in the ICON version described by Giorgetta et al. (2018).

In addition to the distribution of greenhouse gases (GHGs) and meteorological data, the radiation scheme requires input
regarding cloud optical properties, aerosol optical properties and the surface albedo (see e.g. Dietmüller et al., 2016). For a





typical simulation this information now comes from the MESSy submodels CLOUDOPT, AEROPT and the new submodel ALBEDO. Below, we describe for such a typical simulation how these radiation-related submodels (or previous Fortran routines in the case of ALBEDO) have been modified during the revision of the radiation infrastructure. However, we note that it is also possible to feed the respective input, e.g. from a previous simulation, into the RAD submodel via the MESSy submodel

IMPORT (Kerkweg and Jöckel, 2015), which allows among others to read time series of gridded data from netcdf files.

## 2.3    AEROPT: updates of the MESSy submodel for the calculation of aerosol optical properties

The AEROPT submodel (Dietmüller et al., 2016) calculates the aerosol optical properties that are required for the radiative transfer calculation in the RAD submodel, namely: aerosol optical depth for the LW and SW, and single scattering albedo and asymmetry factor for the SW only, as scattering in the LW is neither considered in E5rad (Roeckner et al., 2003), nor in PSrad

(Pincus and Stevens, 2013). These optical properties are wavelength dependent. As the number of SW bands is different for PSrad compared to the old (ECHAM5) radiation scheme, the AEROPT submodel had to be revised. Consequently, the number of wavelength bands can vary between different sets of aerosol optical properties. We achieve this, as now for each call the AEROPT submodel provides CHANNEL objects with the corresponding number of wavelength bands.

Further, the Max-Planck-Institute Aerosol Climatology version 1 (MACv1) for tropospheric aerosol optical properties described by Kinne et al. (2013) was made available via IMPORT and by introducing an ICON (version 2.4.0) routine (new MESSy Fortran module "messy_aeropt_kinne.f90"), which maps the aerosol optical properties to the model's current height profile and merges the climatologies for fine and coarse mode aerosol in the SW (see Giorgetta et al., 2013a, for the mapping and merging details).


All other features of the AEROPT submodel as described by Dietmüller et al. (2016) remain fully functional, e.g. multiple diagnostic calls of the AEROPT submodel or the combination of different aerosol sets. The latter is typically used to merge tropospheric and stratospheric aerosol data and while merging, the consistency of the number of wavelength bands is checked. While it is still available for the old radiation scheme, the coupling of online calculated aerosol is not yet implemented for the

PSrad scheme. However, this functionality is supposed to follow with a revision of the AEROPT submodel.

## 2.4    CLOUDOPT: updates of the MESSy submodel for the calculation of cloud optical properties

The submodel CLOUDOPT (Dietmüller et al., 2016) provides the cloud optical properties, which are needed for the calculation of the radiation in the submodel RAD. So far, in analogy to the aerosol optical properties provided by AEROPT, CLOUDOPT

provides the band-dependent cloud optical properties of optical depth (again for LW and SW), single scattering albedo (SW) and asymmetry factor (SW). We revised the CLOUDOPT submodel to account for the band structure of the new radiation scheme. CLOUDOPT now also contains the calculation of cloud optical properties, as described by Stevens et al. (2013) and implemented in ICON (version 2.4.0). As for the AEROPT submodel, we generalized the infrastructure. Now, the number of





wavelength bands of the CHANNEL objects can vary with each call of the CLOUDOPT submodel. Together with the adaptions
in AEROPT, this allows to call radiation schemes with different spectral resolutions within a single simulation for diagnostic
purposes.

In the LW the mass extinction coefficients of the new scheme follow the ECHAM5 parametrizations (Stevens et al., 2013),
which were presented by Roeckner et al. (2003). For liquid clouds the relation between effective radii and mass extinction is
given in equations 8 and 11.61 of Stevens et al. (2013) and Roeckner et al. (2003), respectively. For ice clouds, the parametriza-
tion is based on Ebert and Curry (1992; see Roeckner et al., 2003; Stevens et al., 2013). In addition, CLOUDOPT also allows
the use of an alternative calculation for ice mass extinction in the LW, which was adopted from ECHAM4 (Eq. 101 and Table 3
of Roeckner et al., 1996). For the SW the new scheme derives the mass extinction, single scattering albedo and asymmetry
factors from look-up tables (Stevens et al., 2013), whereas the old scheme uses a set of coefficients to derive SW optical prop-
erties from effective radii (Roeckner et al., 2003).

As in ECHAM5 and ECHAM6, the cloud optical depths of liquid and ice clouds are rescaled using a cloud inhomogeneity
factor to account for the subgrid-scale variability of clouds (Roeckner et al., 2003; Mauritsen et al., 2012; Stevens et al., 2013;
Mauritsen et al., 2019; Mauritsen and Roeckner, 2020, see keywords "zinhoml" and "zinhomi" in the supporting information
of the latter). For liquid clouds, the inhomogeneity factors can now be set depending on the cloud type (convection type). In
the namelist three inhomogeneity factors can be set for convection-free, convective and certain shallow convective clouds (see
Mauritsen et al., 2019; Mauritsen and Roeckner, 2020, supporting information of the latter) in analogy to the implementation
in ECHAM6.3 and ICON.

## 2.5 ALBEDO: introduction of the new MESSy submodel for the calculation of surface albedos

As a final step to separate code from the RAD submodel that is independent of the radiation scheme, the calculation of the
surface albedo was modularized. Therefore, we introduced the new submodel ALBEDO. This new MESSy submodel contains
the previous (ECHAM5-based) routines to calculate the surface albedo and was extended by adding new parametrizations
and additional features for the calculation of solar zenith angle (SZA) dependent surface albedos. In particular, ALBEDO can
calculate a blue-sky albedo from the black-sky and white-sky albedos and the fraction of direct and diffuse surface radiation
fluxes (see e.g. Li et al., 2018; Liu et al., 2009; Cordero et al., 2021, and references therein for details on the different albedos
and how to typically derive the blue-sky albedo). Further details on the modularization and updates are described below.

**ECHAM5 (background) albedo**

ECHAM5 uses a so-called background albedo for snow-free land surfaces (Roeckner et al., 2003). This temporally constant
(i.e., without interannual or subseasonal variation) climatological field is based on Hagemann (2002). This background albedo



is modified according to meteorological and land properties and an albedo for grid points containing sea ice is calculated (Roeckner et al., 2003). Finally, the resulting fields are combined with a constant value for the albedo of ice-free ocean surfaces to produce the final (blue-sky) albedo, employed in the ECHAM5 model (Roeckner et al., 2003). The corresponding routine is
shifted to the core layer of the new ALBEDO submodel and is called from the respective submodel interface layer.

**New white-sky albedo for snow-free land**

Here, we introduce a new white-sky albedo for snow-free land surfaces, which can be used to calculate SZA dependent surface albedos and is practically a substitute for the previous ECHAM5 background albedo. This white-sky albedo is a monthly mean climatology based on data from the Moderate Resolution Imaging Spectroradiometer (MODIS; https://modis.gsfc.nasa.gov/
about/ accessed last 03 February 2023). Furthermore, in principle, it is possible to use any (background or white-sky) albedo with any temporal resolution as input via IMPORT, since the (background or white-sky) albedo is now namelist controlled. So, besides the newly added monthly climatology with subseasonal variation also other albedo data with different variability (e.g. transient) could be fed in as background albedo via IMPORT.

The provided white-sky albedo was produced from the MODIS/Terra+Aqua BRDF/Albedo Gap-Filled Snow-Free Daily L3 Global 30ArcSec CMG V006 data product (MCD43GFv006; Sun et al., 2017; Schaaf, 2019). We used the white-sky albedo near shortwave broadband and the period from 01 January 2001 to 31 December 2010. The original data are daily files on a 43200 x 21600 grid. This grid corresponds roughly to a pixel size of 1 km x 1 km. Values of the white-sky albedo below 0.07 in the raw daily files are set to missing (guided by the reference value for the ocean surface albedo used in ECHAM5;
Roeckner et al., 2003) and the resulting files are further used to calculate monthly means. We calculate a climatology over all months, which we use to fill in missing values in the original monthly mean files: i.e. we substitute missing values in the original monthly files with a climatological value calculated from the original monthly files where the particular pixel is not missing. Consequently, a 12-month climatology is calculated from the collection of the updated monthly files. The all-time climatology is used to create common generic conversion weights to remap both climatologies (all time and 12 months) to a
$360 \times 180$ grid. Still missing grid points in the two climatological files - which can occur as there might be grid points which are missing in all months, which were used to calculate the climatology - are filled using a nearest neighbour method. This procedure ensures that when the resolution-dependent land mask is applied in a simulation, the white-sky albedo for snow-free land includes land albedo values only.

**Solar zenith angle dependent albedo**

One main aspect during the modularization of the ALBEDO submodel was to include the SZA dependence of the albedo for water, land and snow. For the SZA dependence of the ocean surface, the parametrization as described in Appendix A of Li et al. (2006) was implemented (with a scaling factor to achieve improved global mean SW fluxes; see Sect. 3.2). Li et al. (2006) refer to this parametrization as being based on the Preisendorfer and Mobley (1986) scheme. The SZA-dependent land surface albedo is parametrized depending on the surface properties as in Appendix B of Briegleb (1992; analogous to the im-



plementation in the ICON module mo_albedo.f90). For the snow albedo, we use the parametrization as given in Formula A3
of Yang et al. (2001; see also Appendix B of Briegleb, 1992, and references therein).

When the SZA dependence is used, the procedure to calculate the blue-sky albedos is as follows: The white-sky albedo,
e.g. from MODIS (see above), is modified according to meteorological properties and land properties as well as ice cover
(as was the ECHAM5 background albedo before) and an albedo for sea ice is calculated (again as in ECHAM5). Based on
this white-sky albedo and the respective parametrizations (see previous paragraph) a SZA dependent black-sky albedo for
land (snow-covered and snow-free) and ice-covered (snow-covered and snow-free) surfaces is calculated. Additionally, over
(ice-free) ocean surfaces a white-sky and black-sky albedo is calculated based on the wind speed and the SZA (Yang et al.,
2001). From these white-sky and black-sky albedos and the diffuse and direct SW surface radiation fluxes the blue-sky albedo
is obtained.

To be able to use this new feature, either the radiation scheme has to provide the (fraction of) the direct and diffuse SW
radiation fluxes or the user has to set a fixed relation between these fluxes via namelist. The former is the case for both PSrad
and the SW scheme rad_short_v2, which was slightly adapted to this end, whereas the latter is the case for rad_short_v1.


## 2.6 Minor modifications of the radiation infrastructure

During the restructuring of the radiation infrastructure we made several minor adjustments in addition:

(1): ECHAM5 commonly performs (full) radiation calls less frequently than at each model time step (Roeckner et al., 2003).
Thus output from a specific radiation call is used for several model time steps. Hence, at a time step when a new (full) radiation
call is performed, the orbital parameters are advanced (by $\Delta t_{orb}$) for the radiation call (Roeckner et al., 2003). The results
from this radiation call (with the adjusted orbital parameters) are later on adjusted with the orbital parameters of the actual
model time step for the calculation of the actual SW fluxes and heating rates (see Roeckner et al., 2003). Figure 2 illustrates the
alignment of model time steps and radiation calls. Previously, the orbital parameters were shifted to the middle of the interval
between the current and the next full radiation call, including the latter (Fig. 2a). Now, the offset type can be selected via a new
namelist switch. Apart from the previous choice $\Delta t_{orb,opt0}$, the orbital parameters now can be chosen to be calculated for the
middle of the interval of time steps associated with the current radiation call ($t_{r,i-1}$, $t_{r,i-1} + \Delta t_m$,...., $t_{r,i} - \Delta t_m$, leading to
$\Delta t_{orb,opt1} = \frac{1}{2}((t_{r,i} - \Delta t_m) - t_{r,i-1})$, Fig. 2b), or the offset can be set to an arbitrary constant ($\Delta t_{orb,con} \leq \Delta t_r$).

(2): The calculation of the dry air column and the corresponding water vapour, which are passed to the core radiation scheme,
were slightly adjusted. In short test simulations we found the effect of the changes to be only of minor importance. As this
change had been implemented before the radiation infrastructure was updated, it also applies to the simulations with E5rad.
However, for the sake of completeness, we mention it here.

(3): The so-called diffusivity factor (see e.g. Roeckner et al., 2003; Li, 2000, and references in the latter), which is used to
scale the optical thickness of the clouds in the LW, was removed from the CLOUDOPT submodel and is now accounted for





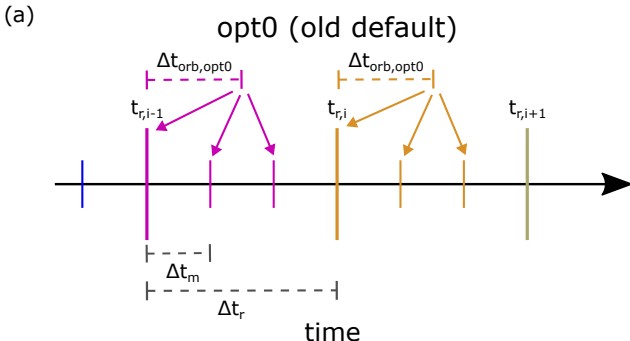

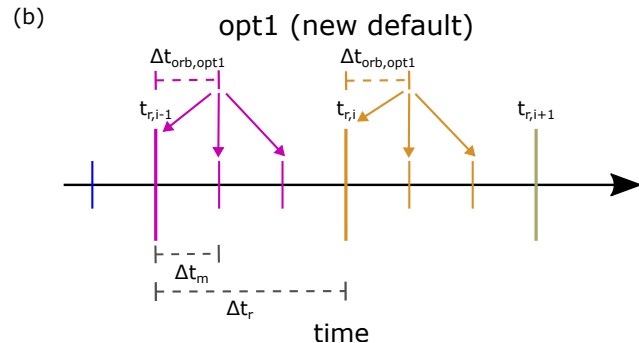

**Figure 2.** Schematic of the radiation calls for 3 model time steps per full radiation call (long vertical lines, e.g. $t_{r,i}$) for the old (a) and new (b) choice of the offset parameter ($\Delta t_{orb}$): For $\Delta t_m < \Delta t_r$ (no full radiation calculation at every model time step), previously (a) the orbital parameters where shifted according to $\Delta t_{orb,opt0} = \frac{1}{2}(t_{r,i} - t_{r,i-1})$, whereas the new option shifts the parameter according to $\Delta t_{orb,opt1} = \frac{1}{2}((t_{r,i} - \Delta t_m) - t_{r,i-1})$. In addition to the old and new choice of the offset parameter ($\Delta t_{orb}$), it is now also possible to set this parameter via namelist to a constant ($\Delta t_{orb,con} \leq \Delta t_r$).

(exactly once) in the radiation schemes to avoid any confusion. Originally, the application of the diffusivity factor was partly

mixed into the parameters that were used to calculate LW cloud optical thicknesses and partly applied later in the code for the new radiation scheme, while it was accounted for in the cloud optical properties for the old scheme. This restructuring caused changes in the output of CLOUDOPT and the binary divergence of model results based on the old and the new code when applying the old (ECHAM5) radiation scheme.

     (4): The distance between Sun and Earth (zdisse) was updated to account for the shift of the orbital parameters by $\Delta t_{orb}$.

Although this change is expected to have a negligible impact on the model results, we note it here, as it destroys binary identity.

### 2.7    Overview of the new radiation infrastructure dependencies

The interplay of the radiation-related submodels is presented as a schematic in Fig. 3 for a typical (new) setup. Red arrows mark the two new dependencies that now exist: 1) The direct and diffuse surface fluxes from the last radiation update (box "rad upd." in Fig. 3) are provided to the ALBEDO submodel. 2) The orbital parameters (most importantly the SZA) are calculated





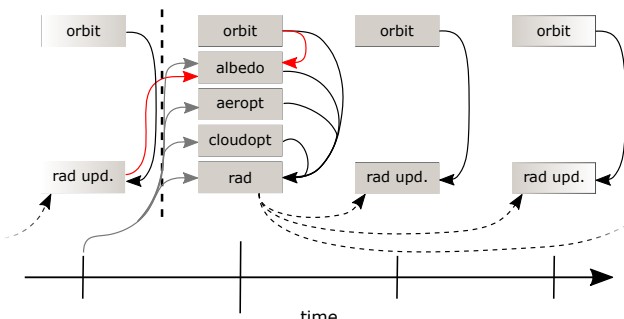

**Figure 3.** Schematic of the interdependencies of the radiation infrastructure for a typical simulation setup with the new radiation scheme. Grey arrows show information (e.g. temperature, pressure) from the model time step (vertical bars) before a full radiation time step (long vertical bar) that is passed into radiation-related submodels. In a full radiation step (long vertical bar), the radiative transfer is calculated and stored. Dashed arrows: information from a full radiation time step is forwarded to radiation update (rad upd.) time steps. At these time steps for the SW, updates of the radiative fluxes and heating rates are calculated and applied (see Roeckner et al., 2003). Red arrows show new dependencies: Input from the previous radiation update (fluxes at surface) and the information from ORBIT (mainly SZA) are fed to the ALBEDO submodel.

by ORBIT and provided to the ALBEDO submodel, which then calculates the albedo for the next full radiation calculation. We note that the latter dependency was hidden before as the calculation of the surface albedo was performed in the RAD submodel. While the other dependencies (black arrows) have already existed before our developments, all submodels (RAD, ALBEDO, CLOUDOPT, AEROPT) except for ORBIT have been revised and are more flexible now.

The processing chain of the radiation calculation is as follows: At a full radiation time step (long vertical bar in Fig. 3), the information (e.g. temperature, pressure, cloud, aerosol, gases, ...) from the previous model time step is available to ALBEDO, AEROPT, CLOUDOPT and RAD. Additionally, fluxes from the last radiation update are available for the ALBEDO submodel, which also receives information from ORBIT, in particular the SZA. Then, the different radiation related submodels are called and pass their information to RAD. Finally, the full radiation calculation is performed with an offset of the orbital parameters
and the results are stored. The SW fluxes at the model time steps are then calculated via a simple update of the radiation fluxes (as in ECHAM5 see Roeckner et al., 2003). Note that "rad upd." is also performed for the full radiation time step as the orbital parameters used for the radiative transfer are typically shifted in comparison to the orbital parameters (mainly SZA) associated with the current model time step (see $\Delta t_{orb}$ Sect. 2.6).





## 3 Evaluation of the new (dynamic) configuration

During the implementation of the presented updates, it was ensured that previous model results could be reproduced after the restructuring of the code. In particular, binary identity was secured up to a point, where required changes (see Sect. 2.6, e.g. "diffusivity factor") break binary identity. A key strength of the MESSy concept is that many (including previous) model configurations can be run with the same executable by adjusting Fortran namelists only. Accordingly, the four simulations discussed hereafter can be performed with the same executable by changing three namelist files (RAD, ALBEDO and IMPORT)

only. As we have performed diagnostic radiation calls with an exchanged radiation scheme (e.g. driving the simulation with PSrad and performing an additional diagnostic radiation call with E5rad; see Sect. 4), the CLOUDOPT and AEROPT namelist files already included the calculation of aerosol and cloud optical properties for both (E5rad and PSrad) radiation schemes. Hence, these namelist files did not have to be adjusted when the driving radiation scheme was switched.

### 3.1 Simulation setups

We performed four simulations for the evaluation presented here. Namely, two simulations (pre-industrial and present-day denoted with pi and pd, respectively) for each of the two radiation schemes (the old ECHAM5 radiation scheme with the v2 in the SW, denoted here with E5rad, and the newly implemented PSrad scheme). These simulations will be addressed here as EMAC-E5rad-pi, EMAC-E5rad-pd, EMAC-PSrad-pi and EMAC-PSrad-pd, respectively.

The simulations were conducted with T42 spectral truncation and 90 vertical levels extending up to roughly 80 km (see the T42L90MA setup e.g. mentioned by Jöckel et al., 2016). For the solar forcing we applied a total solar irradiance (TSI) of 1360.75 Wm$^{-2}$, representing approximately the average TSI of the first two decades (first two solar cycles) in the time series displayed in Fig. 1 of Matthes et al. (2017a; Matthes et al., 2017b; data also available from https://solarisheppa.geomar.de/ cmip6), i.e. representing pre-industrial conditions. Although Fig. 1 of Matthes et al. (2017a) indicates an increase in TSI from

the pre-industrial conditions to the end of the 20th century (to roughly 1361.25 Wm$^{-2}$), we have kept the TSI constant for the present-day simulations. The increase of about 0.5 Wm$^{-2}$, is not of substantial relevance in the global energy budget of Earth, as only 1/4 of this difference remains for Earth's global average, which is further reduced as a part of this additional solar irradiance is reflected. Thus we expect the change from pre-industrial to present-day conditions to be in the order of about 0.1 W m$^{-2}$ in the end.


Table 2 presents additional forcings and boundary conditions. These represent pre-industrial (pi, representative of the year 1850 conditions with some deviations due to data availability) and present-day conditions (pd, representative of the year 2000 conditions). A short outline of the employed boundary data is given below.

The four simulations use prescribed sea surface temperatures (SSTs) and sea ice cover (SIC; Rayner et al., 2003) and the quasi-biennial oscillation (QBO) is nudged as described by Jöckel et al. (2016). Except for simplified methane chem-



istry, these simulations are purely dynamic. In the lowest model level methane (CH$_4$) is nudged to surface mixing ratios according to historical CMIP6 data (Meinshausen et al., 2017). In the atmosphere the simplified methane chemistry includes two effects: (i) The methane oxidation, which is represented by the MESSy submodel CH4 (Winterstein and Jöckel,

2021) using prescribed climatologies of the methane reactions partners (OH, O($^1$D), Cl) from previous EMAC simulations: EMAC-DECK-piControl and EMAC-RD1-base-01 (Jöckel, 2023, see also https://data.ceda.ac.uk/badc/ccmi/data/post-cmip6/ ccmi-2022/DLR/EMAC-CCMI2/refD1), which were conducted according to the CMIP6 (Eyring et al., 2016) and CCMI-2 (https://blogs.reading.ac.uk/ccmi/ccmi-phase-two/, accessed last 17 July 2023; for phase one of CCMI see Eyring et al., 2013; Morgenstern et al., 2017) protocols. Water vapour tendencies due to methane oxidation are consequently accounted for in the

interactive water vapour field of the simulation. (ii) Methane is photolyzed using a photolysis rate which is calculated online by the MESSy submodel JVAL (Sander et al., 2014). The corresponding water vapour and methane fields are used in the first call of the radiation module and thus are driving the simulation.

All other trace gas fields required by the radiation schemes, e.g. carbon dioxide (CO$_2$), nitrous oxide (N$_2$O), ozone (O$_3$)

and the chlorofluorocarbons CFC-11 and CFC-12, also stem from comprehensive chemistry-climate model simulations, which were previously conducted with EMAC, namely EMAC-DECK-piControl and EMAC-RD1-base-01. Additional diagnostic radiation calls were performed with the imported methane fields from these previous EMAC simulations.

The CO$_2$, CH$_4$ and N$_2$O fields of these previous simulations in turn are based on the respective historical CMIP6 data

presented by Meinshausen et al. (2017), which are used as lower boundary conditions in these simulations. Table 1 presents the climatological surface level mixing ratios of these simulations. These values are in agreement with the values presented in Table 5 of Meinshausen et al. (2017) for 1850 and 2000 conditions.

**Table 1.** Global mean surface level (lowest model level) mixing ratios as employed in the pi and pd simulations based on fields from the previous EMAC simulations EMAC-DECK-piControl and EMAC-RD1-base-01.

|  | CO$_2$ ($\mu$mol mol$^{-1}$) | N$_2$O (nmol mol$^{-1}$) | CH$_4$ (nmol mol$^{-1}$) | CFC-12 (pmol mol$^{-1}$) | CFC-11 equiv. (pmol mol$^{-1}$) |
|---|---|---|---|---|---|
| pi | 284.3 | 272.9 | 804.7 | 0 | 2.4 |
| pd | 368.9 | 315.0 | 1760 | 528.7 | 492.8 |

For CFC-12 the global mean values in the lowest model level are 0 and 528.7 pmol mol$^{-1}$ for pi and pd conditions, respec-

tively (see Tab. 1). These values are in agreement with the lower boundary values they are based on, which were presented by Meinshausen et al. (2017) and Carpenter et al. (2018). To include the effect of additional radiatively active ozone-depleting substances (ODSs), the approach outlined by Meinshausen et al. (2017) to lump additional radiatively active ODSs via radiative efficiencies (see e.g. Burkholder, 2018) to CFC-11 equivalents for purposes of radiative transfer calculations was applied in



the EMAC-DECK-piControl and the EMAC-RD1-base-01 simulations. For the EMAC-DECK-piControl, eight species have been

been lumped to CFC-11 equivalents based on values presented by Meinshausen et al. (2017), whereas for the EMAC-RD1-base-01 only six species have been lumped according to the data given by Carpenter et al. (2018). This results in global mean values of 2.4 pmol mol$^{-1}$ and 492.8 pmol mol$^{-1}$ of CFC-11 equivalents in the lowest level of the EMAC-DECK-piControl and EMAC-RD1-base-01 simulation, respectively. This is lower than the expected full CFC-11 equivalents for the respective periods, which are in the order of 30 pmol mol$^{-1}$ for pre-industrial conditions and above 700 pmol mol$^{-1}$ for the 2000s (see

Meinshausen et al., 2017). However, the lower CFC-11 equivalent mixing ratios in the EMAC simulations, are in agreement with the respective reference values given the reduced number of accounted (lumped) species in the model setups.

In all simulations stratospheric aerosol data from the ETH Zürich (ETHZ) (2017), as proposed for CMIP6, were employed. The tropospheric aerosol data is based on Tanre et al. (1984) and Kinne et al. (2013) for E5rad (as described by Roeckner et al.,

2003, for ECHAM5) and PSrad, respectively. Concerning the surface albedo, the E5rad simulations use the previous ECHAM5 routines to adapt the ECHAM5 background albedo (for details see Hagemann, 2002; Roeckner et al., 2003), whereas the PSrad simulations use the surface albedo computed with the newly implemented solar zenith angle dependent albedo (for water, land and snow), where the white-sky albedo for snow-free land was derived from MODIS (see Sect. 2.5). Hence, except for tropospheric aerosol data and the albedo, the boundary conditions for the E5rad and PSrad simulations were identical.


After optimizing the set of free parameters of the model with respect to the boundary data and the respective radiation scheme (see description in Sect. 3.2), the simulations have been performed for 20 years, while our analyses exploit only the last 10 years of each of the simulations to reduce the risk of any possible influence from the spin-up period. To reduce the amount of data, model output was aggregated as monthly mean values on model levels. These monthly means were calculated online

(i.e. all model time steps are accounted for in the means) and, whenever necessary, they were interpolated to pressure levels offline.

## 3.2 Parameter optimization for the dynamic model setups

Earth receives approximately 0.34 kW m$^{-2}$ of solar radiation at the top of the atmosphere (TOA) on average, which is almost

balanced by TOA reflected SW radiation ($\sim 0.1$ kW m$^{-2}$) and TOA outgoing LW radiation ($\sim 0.24$ kW m$^{-2}$; e.g. Trenberth et al., 2009; Stephens et al., 2012; Wild et al., 2015). It is challenging to assess the resulting imbalance (Johnson et al., 2016), which is somewhat below 1 W m$^{-2}$ (e.g. Trenberth et al., 2009; Wild et al., 2015; Johnson et al., 2016, which present estimates within 0.6–0.9 W/m$^2$). The best estimates are derived from heat uptake analyses (Johnson et al., 2016), which are used to calibrate satellite-based observations (Loeb et al., 2009, 2018).


Similarly, in global (climate) models the TOA (im)balance is commonly "calibrated" to observed estimates during the so-called tuning process (Hourdin et al., 2017). Here, we optimize the four setups that are described in the section above (Sect. 3.1).



Our two primary targets were (i) a radiative balance at TOA close to $0\,\mathrm{Wm^{-2}}$ for the pre-industrial configuration (assuming that during that period the Earth's energy budget was almost balanced) and (ii) a radiative imbalance at TOA around $1\,\mathrm{Wm^{-2}}$

for the present-day configuration with the same parameter set (accounting for the expected imbalance; see above). Further, we aimed for clear- and all-sky LW and SW present-day TOA radiation fluxes to be within the uncertainty range of satellite-based observational estimates (Loeb et al., 2018; CERES Science Team, 2021), while securing the hydrological cycle to remain within an acceptable range compared to observations (see below). For a more elaborate review of the principles of climate model tuning, which we will address also as parameter optimization in the following, we refer the reader to Mauritsen et al.

425 (2012).

To achieve our goals, we adjusted parameters associated with clouds, convection and the surface albedo, while keeping the previous defaults e.g. for parameters related to the parametrization of gravity waves. Table 3 lists the final parameter set along with previously used parameter values. Prior knowledge of sensitivities of the radiative fluxes regarding typical optimization

parameters from Mauritsen et al. (2012; Fig 3) and Kern (2013; Appendix D) allowed us to adjust parameters target-oriented without extensive testing of all possible sensitivities.

As a starting point for the model optimization, we used typical ECHAM6.3 values for the inhomogeneity factors for liquid and ice clouds (Mauritsen et al., 2019). All other optimization parameters were set to the previous EMAC defaults. Firstly,

we targeted the TOA global annual mean clear-sky SW fluxes via the surface albedo as there is no (substantial) dependence of these fluxes on the other optimization parameters. During this process, we increased the minimum albedo of bare sea ice from 0.50 (see Roeckner et al., 2003) to 0.55 (a value that has been previously used in other EMAC simulation setups) and increased the ocean surface albedo by a factor of 1.15 to enhance the outgoing SW clear-sky radiation at TOA to roughly match satellite-based estimates (Loeb et al., 2018). Secondly, we targeted the TOA LW flux by increasing a parameter that influences

a geopotential-based conversion rate from cloud water to rain in convective clouds (cprcon) to the value used for ECHAM6.3 in T63 spectral resolution (Müller et al., 2018). Thirdly, targeting the TOA SW flux, which is sensitive towards various parameters (see e.g. Mauritsen et al., 2012), we decreased the convective mass flux above the level of non-buoyancy (cmfctop) to 0.23, which now lies between the previous EMAC default and the value used in ECHAM6.3 in T63 spectral resolution (Müller et al., 2018).

Figure 4 shows various radiation fluxes along with reference values from observations (Loeb et al., 2018; CERES Science Team, 2021) and results from CMIP6 (Wild, 2017). Both, the observations and the CMIP6 results in Wild (2017) are representative of present-day conditions. The global mean radiation (im)balance in the EMAC simulations is somewhat above $1\,\mathrm{W\,m^{-2}}$ for present-day conditions and somewhat below $0\,\mathrm{W\,m^{-2}}$ for pre-industrial conditions with slightly more deviation from the target values for the E5rad simulations. The absolute values of the LW and SW all-sky fluxes are slightly too low on average

in the EMAC simulations compared to observational data. Overall the various fluxes from the optimized simulations lie close to or within the uncertainty range of observations.



### 3.3 Comparison of old and new model configuration

After optimizing the model configurations for pre-industrial and present-day conditions, we compare the climatological mean
states of key meteorological quantities with reanalysis and observational data. For the reanalysis data we employ ERA5 (Hersbach et al., 2020) monthly mean data on pressure levels (Hersbach et al., 2023) obtained from Copernicus Climate Change Service, Climate Data Store (2023). The model data was interpolated vertically to the pressure levels of the reanalysis (pressure level) data set, whereas the ERA5 data was horizontally regridded to the T42 resolution of the model data. For the evaluation of simulated precipitation data, we use the monthly mean observational data from the Global Precipitation Climatology Project (GPCP, e.g. Huffman et al., 1997, 2009; Adler et al., 2003) version 2.3 (Adler et al., 2018). For both reanalysis and observational data we use the period 2000–2009 for intercomparison with the last ten years of our simulations (see Sect. 3.1).

Figure 5 shows the differences in the zonal mean temperatures between the model present-day configurations and ERA5 (first two columns) and between the two present-day simulations with different driving radiation schemes (PSrad and E5rad, third column). Up to around 30 hPa, both model configurations show similar bias patterns compared to ERA5. These biases tend to be lower for EMAC-PSrad-pd than for EMAC-E5rad-pd, except for the extratropical stratosphere in the height region between 150 and 30 hPa. Above 30 hPa EMAC-PSrad-pd shows mostly higher temperatures than EMAC-E5rad-pd. Hence, where E5rad was on average too cold in the region above 30 hPa the EMAC-PSrad-pd simulation results seem to be too warm in comparison with ERA5 data and the warm bias at 60-40° S during JJA compared to ERA5 is even more pronounced in EMAC-PSrad-pd. However, in large regions EMAC-PSrad-pd performs better e.g. concerning the cold bias around the tropical cold point (which is reduced by about 3 K) and the reduced cold bias in the extratropical lower stratosphere.

The cold bias in the tropical upper troposphere and lower stratosphere, as well as other biases of EMAC-E5rad-pd compared to ERA5 are similar to what has been found by Jöckel et al. (2016) when comparing annual climatologies of EMAC simulations with ERA-Interim data (see their Fig. 12; in particular the panel for the RC1-base-01 simulation). Previous comparisons of ECHAM5 and ERA-Interim data for DJF presented by Stevens et al. (2013) show similar biases as our EMAC-E5rad-pd simulation (see their Fig. 12). Stevens et al. (2013) also find a resolution-dependent warming and a reduction of the cold biases during DJF when ECHAM6.1 (including an updated radiation scheme compared to ECHAM5) is employed. These changes from ECHAM5 to ECHAM6.1 are similar to the improvements we have found when assessing EMAC-PSrad-pd compared to EMAC-E5rad-pd simulations.

Figure 6 shows the corresponding zonal mean zonal wind differences. The main biases between the model data and ERA5 remain unchanged when the newly available radiation scheme, PSrad, is used. These biases have already been present in comparisons of ERA-Interim data with ECHAM5 and ECHAM6.1 data (Stevens et al., 2013, see their Fig. 13). EMAC-PSrad-pd shows reduced biases at 60° S in comparison to EMAC-E5rad-pd. However, in the SH polar region during JJA above 50 hPa the positive bias is increased in EMAC-PSrad-pd. In the tropical upper troposphere eastward winds are present in EMAC-E5rad-





pd, whereas ERA5 shows westward winds in this region. This bias slightly increases in the simulation with PSrad. Differences between EMAC-E5rad-pd and EMAC-PSrad-pd show increased wind speeds during JJA in the SH polar vortex (Fig. 6i). This strengthening of the polar vortex is desirable as the polar vortex in EMAC is known to be too weak (Jöckel et al., 2016).

490

Although the analyses only include the last 10 years of both the E5rad-pd and the PSrad-pd simulation, the results from the pre-industrial simulations support the general features presented here. In particular, the patterns of the differences that arise when employing the new radiation scheme (PSrad) and the previous ECHAM5 scheme (E5rad) are similar under present-day and pre-industrial conditions.

495

Figure 7 shows specific humidity profiles (in kg per kg of moist air) for different latitudinal bands from the tropics to the high latitudes. Overall, all data sets show the typical decrease of specific humidity with height in the troposphere. Above approximately 100 hPa, ERA5 shows higher specific humidity than the model data. At this altitude, the EMAC-PSrad-pd simulation is moister (and thus in better agreement with ERA5) than the EMAC-E5rad-pd simulation, which is consistent with higher tropical cold point temperatures in the EMAC-PSrad-pd simulation compared to the EMAC-E5rad-pd simulation (see Fig. 5). In general, ERA5 reaches the low stratospheric humidity values somewhat below (at higher pressures than) the model data. This is particularly obvious in the NH and SH polar cap profiles, where in the height region near 200 hPa ERA5 has already reached minimum specific humidity values in the range of $2\text{-}3\times10^{-6}\,\mathrm{kg\,kg^{-1}}$ and the EMAC simulations still show a roughly linear decrease in specific humidity (in log-log) up to somewhat below 100 hPa. Due to a slower decrease and a slight kink in the specific humidity profiles over the polar cap regions in the EMAC simulations, specific humidity values are higher in the EMAC simulations than in ERA5 around 200 hPa over the polar caps. After reaching the minimum specific humidity values in the upper troposphere–lower stratosphere region, the specific humidity values increase slightly with height. We attribute this increase to the moistening through methane oxidation, which increases with height up to at least 10 hPa in the model (Eichinger and Jöckel, 2014, see their Fig. 8).

510

Seasonal variations of tropical stratospheric water vapour related to the water vapour tape recorder (Mote et al., 1996) are shown in Fig. 8 for the last 10 years of the EMAC simulations and the period from 2000 to 2009 for ERA5. An intercomparison is feasible due to the selection of the transient SSTs and the nudging of the QBO in the EMAC simulations (see Table 2). The left panels (Fig. 8a and c) show the time series of specific humidity at 70 hPa and 50 hPa averaged over 10° S–10° N. All data sets show a clear seasonal variation and, as noted before, EMAC-PSrad-pd shows higher values than EMAC-E5rad-pd, which are in better agreement with ERA5. The amplitudes of the seasonal cycle of stratospheric water vapour are largest in ERA5 and smallest in the EMAC-E5rad-pd simulation. According to Brinkop et al. (2016), this can be attributed to the too low tropical cold point temperatures in EMAC-E5rad-pd. From comparing panels a) and c) of Fig. 8 the time lag of the water vapour signal propagation is apparent. Further, the amplitudes of the water vapour variations decrease with height in all data sets as can be expected (Mote et al., 1996, 1998).





To assess the amplitude of the variations, Fig. 8 also shows the relative anomalies of specific humidity for the same region (panels b and d). We calculated the anomalies as $(q(t)-\overline{q})/\overline{q}$, where $q(t)$ is the monthly specific humidity value and the overbar denotes the mean (all months weighted equally) of the displayed period. The amplitude and signal strength are captured better in EMAC-PSrad-pd than in EMAC-E5rad-pd when taking ERA5 as a reference. Similar to the absolute amplitudes, the relative amplitudes also decrease with height.

Figure 9 shows the 10-year mean zonal mean precipitation for the model data and GPCP v2.3. Table 4 presents the corresponding tropical ($30°$ S–$30°$ N) and global means. Overall, the largest differences between model and observational data are found in the tropics ($30°$ S–$30°$ N) and in the region $40°$ S–$70°$ S. In the tropics all simulations show enhanced precipitation in comparison to the observational data. On average, the tropical ($30°$ S–$30°$ N) mean precipitation lies between 3.62 and 3.78 mm day$^{-1}$ in the simulations, whereas GPCP v2.3 shows 3.05 mm day$^{-1}$. Further, the different simulation periods of pi and pd seem to have a smaller impact on the precipitation distribution than the exchange of the radiation scheme, i.e. blueish (reddish) lines are more similar than solid (dashed) lines, respectively. The global mean precipitation is 3.00–3.11 mm day$^{-1}$ in the simulations and 2.70 mm day$^{-1}$ in the GPCP v2.3 data. Both, the distribution of simulated precipitation and the global and tropical mean values are comparable to previous EMAC results presented by Jöckel et al. (2016, their Fig. 13), where only EMAC simulations which include global mean temperature nudging showed considerably less precipitation.





**Table 2.** Boundary conditions of the simulations for pre-industrial and present-day conditions with radiation scheme E5rad and PSrad. Monthly mean data is abbreviated via mm. Please see the text for details.

| Data/Forcing | Source (reference) | type |
|---|---|---|
| **pre-industrial: pi** | | |
| SST/SIC | HadISST (Rayner et al., 2003) | mm transient (1875-1894) |
| QBO | FUB (Naujokat, 1986)* | mm transient (1875-1894) |
| $O_3$, OH, Cl, $O(^1D)$, $CH_4$, $CO_2$, $N_2O$, CFC-11 eq., CFC-12 | EMAC-DECK-piControl (CMIP6) | mm climatology from 20 years of 1850 time slice |
| $CH_4$ (call 01)** | Meinshausen and Vogel (2016); Meinshausen et al. (2017) (CMIP6) | mm of year 1850 (cyclic) as lower boundary and CH4 submodel |
| strat. aerosol | ETH Zürich (ETHZ) (2017) (CMIP6) | mm of year 1850 (cyclic) |
| trop. aerosol/E5rad | Tanre et al. (1984) | climatology |
| trop. aerosol/PSrad | Kinne et al. (2013) | mm of year 1865 (cyclic)*** |
| albedo E5rad | ECHAM5 (Roeckner et al., 2003) | based on modified background albedo (Hagemann, 2002) |
| albedo PSrad | MODIS (Sun et al., 2017; Schaaf, 2019) | mm climatological white-sky albedo based on MODIS; SZA dependence |
| **present-day: pd** | | |
| SST/SIC | HadISST (Rayner et al., 2003) | mm transient (1990-2009) |
| QBO | FUB (Naujokat, 1986)* | mm transient (1990-2009) |
| $O_3$, OH, Cl, $O(^1D)$, $CH_4$, $CO_2$, $N_2O$, CFC-11 eq., CFC-12 | EMAC-RD1-base-01 (CCMI-2) | mm climatology from transient run 1990-2009 |
| $CH_4$ (call 01)** | Meinshausen and Vogel (2016); Meinshausen et al. (2017) (CMIP6) | mm of year 2000 (cyclic) as lower boundary and CH4 submodel |
| strat. aerosol | ETH Zürich (ETHZ) (2017) (CMIP6) | mm of year 2000 (cyclic) |
| trop. aerosol/E5rad | Tanre et al. (1984) | climatology |
| trop. aerosol/PSrad | Kinne et al. (2013) | mm of year 2000 (cyclic)*** |
| albedo E5rad | ECHAM5 (Roeckner et al., 2003) | based on modified background albedo (Hagemann, 2002) |
| albedo PSrad | MODIS (Sun et al., 2017; Schaaf, 2019) | mm climatological white-sky albedo based on MODIS; SZA dependence |

*): For the QBO an extension method (see https://www.pa.op.dlr.de/CCMVal/Forcings/qbo_data_ccmval/u_profile_195301-200412.html for a description, last access 19 July 2023) was applied to observational data available from FUB (https://www.geo.fu-berlin.de/en/met/ag/strat/produkte/qbo/index.html, last access last access 19 July 2023; see also Naujokat, 1986).

**) Lower boundary conditions and simplified methane chemistry were used to produce the $CH_4$ field, which drives the simulations. However, for additional radiation calls the $CH_4$ from previous EMAC simulations as for other GHGs, is being used to ensure that the $CH_4$ fields are identical in the simulation driven with E5rad and PSrad and that they match with the other GHGs.

***) The aerosol data set by Kinne et al. (2013) is a mm climatology for the coarse aerosol whereas the fine mode aerosol is mm transient (see also Giorgetta et al., 2013b).



**Table 3.** Comparison of optimized parameters for the final simulation setups with previously used values. Note that the parameter values for the newly optimized simulations (middle column) are within an acceptable range of previously used parameter sets for ECHAM (right column).

| Parameter | EMAC-PSrad/E5rad values | ECHAM reference values |
|---|---|---|
| inhomogeneity factors for liquid clouds (zinhoml)* | 0.80 / 0.40 / 0.80 | 0.80 / 0.40 / 0.80 (Mauritsen et al., 2019)** |
| inhomogeneity factor for ice clouds (zinhomi) | 0.80 | 0.80 (Mauritsen et al., 2019) |
| parameter to influence the geopotential-related conversion rate from cloud water to rain in convective clouds (cprcon in $s^2\,m^{-2}$) | $2.5\times10^{-4}$ | $2.5\times10^{-4}$ (Müller et al., 2018)*** |
| convective mass flux above the level of non-buoyancy (cmfctop) | 0.23 | 0.20 (Müller et al., 2018)*** |
| minimum albedo of bare sea ice (calbmin) | 0.55 | 0.5 (Roeckner et al., 2003) |
| new scaling parameter for the solar zenith angle dependent ocean surface albedo (osapmfac)**** | 1.15 | - |

*) For radiation calls with the old radiation scheme, E5rad, zinhoml is calculated based on total liquid water path an another parameter (zinpar) according to eq. 11.52-11.53 in Roeckner et al. (2003).

**) Mauritsen et al. (2019) only discern certain shallow convective clouds with a different zinhoml factor; this is accounted for by setting two of the three zinhoml parameters to 0.8 in our simulations.                                                                                      .

***) Here, we cite the parameters as listed for MPI-ESM1.2-LR by Müller et al. (2018).

****) Only applicable for simulations driven by PSrad

**Table 4.** Annual mean precipitation ($mm\,day^{-1}$) over the last ten years of the simulations and for 2000–2009 for GPCP_v2.3 data.

| | tropics (30° S–30° N) | global |
|---|---|---|
| EMAC-E5rad-pi | 3.75 | 3.08 |
| EMAC-PSrad-pi | 3.62 | 3.00 |
| EMAC-E5rad-pd | 3.78 | 3.11 |
| EMAC-PSrad-pd | 3.62 | 3.01 |
| GPCP_v2.3 | 3.05 | 2.70 |





**Figure 4.** Radiation fluxes (in W m$^{-2}$) for the pi and pd simulations driven by E5rad and PSrad in comparison to estimates from observational data. The estimates (blue horizontal lines) are based on Loeb et al. (2018) with updates presented by the CERES Science Team (2021). The grey shading marks the respective uncertainties and we aimed for the radiation fluxes (mainly from the pd simulation) to be located within the shaded region after completion of the optimization process. CMIP6 data from Wild (2017) shows the multi-model mean and the inter-model standard deviation.

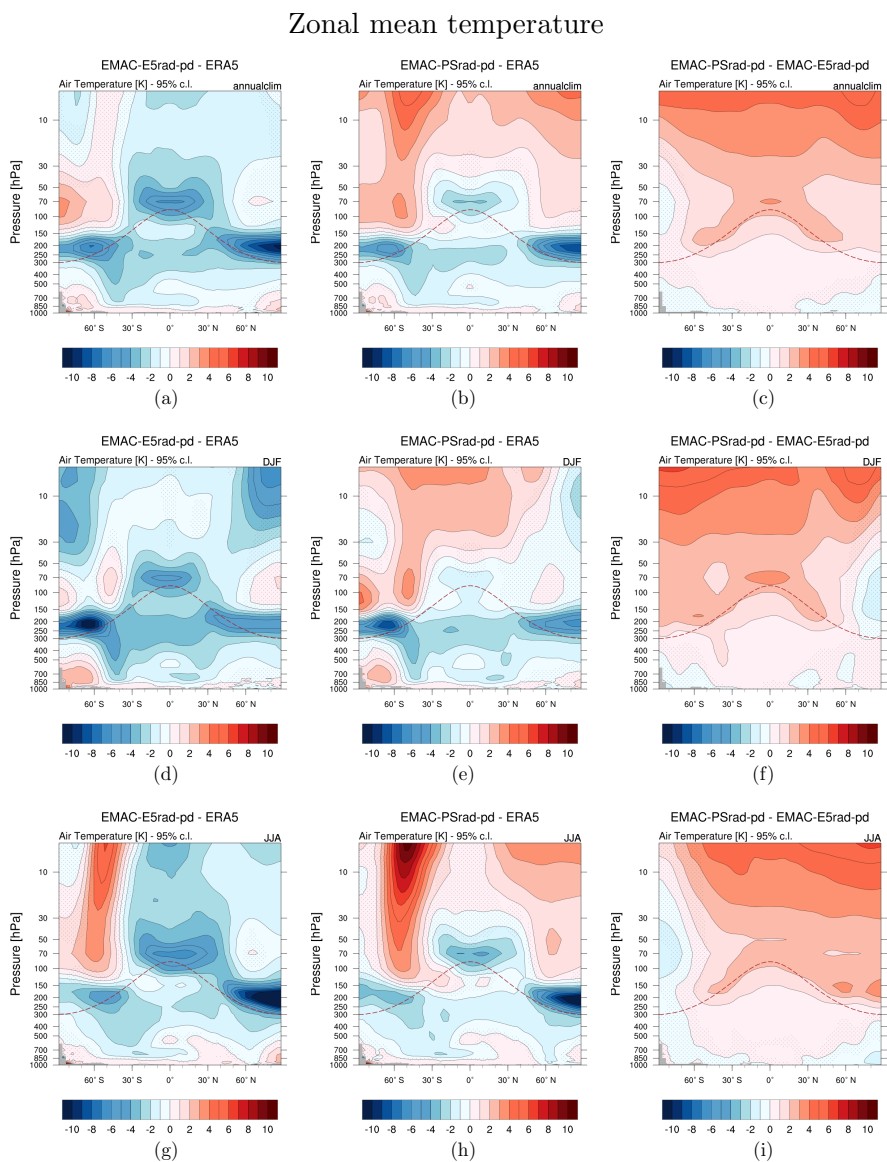

**Figure 5.** Differences of multiannual zonal mean temperatures between EMAC-E5rad-pd and ERA5 **(a, d, g)**, EMAC-PSrad-pd and ERA5 **(b, e, h)**, and the differences between EMAC-E5rad-pd and EMAC-PSrad-pd **(c, f, i)**. Differences in the annual means are shown in the first row, whereas the second and third row show differences for DJF and JJA means, respectively. Stippled regions are not significant on the 95% level based on Welch's t-test. The dashed line indicates a simple latitudinally-dependent approximation of the tropopause (Jöckel et al., 2000).



## Zonal mean zonal wind

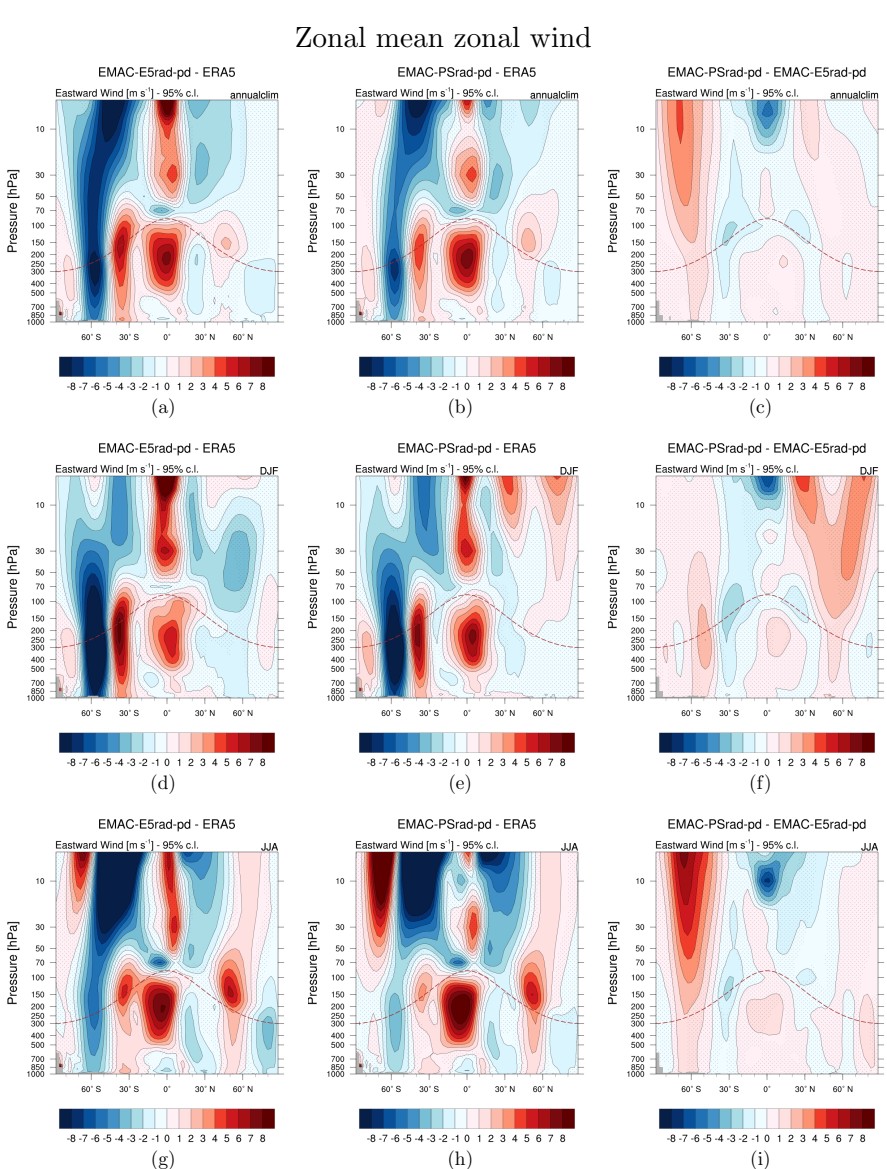

**Figure 6.** As in Fig. 5 but for the differences of multiannual zonal mean of zonal winds.

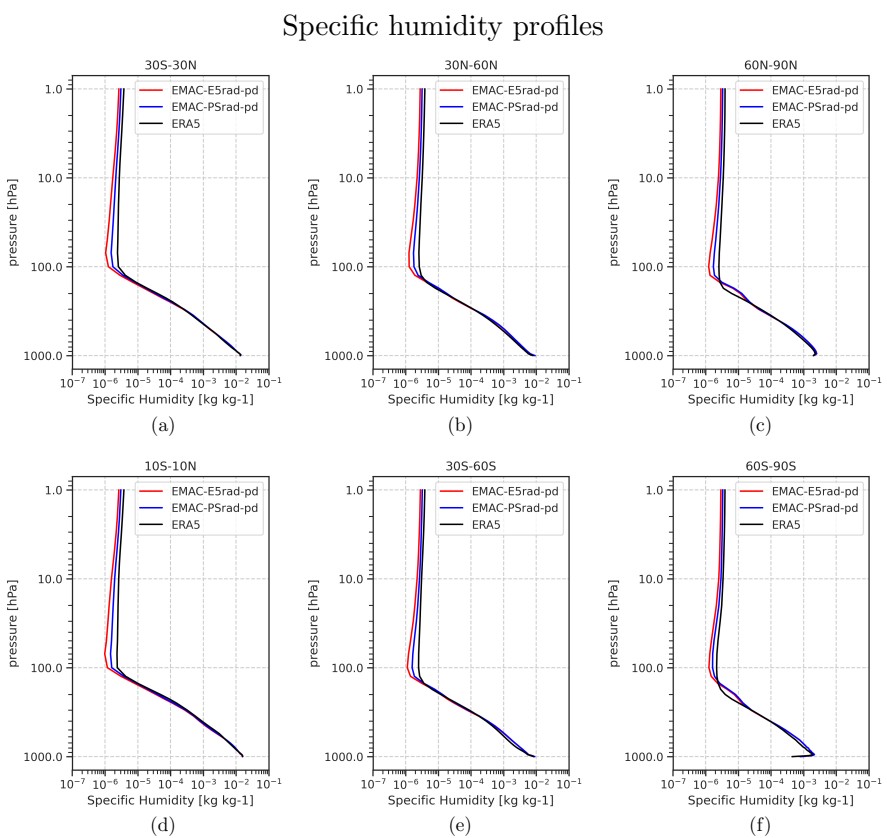

**Figure 7.** Profiles of specific humidity ($\mathrm{kg\,kg^{-1}}$) for various latitudinal bands based on a 10-year climatology. The bands are for the tropics $30^\circ$ N–$30^\circ$ S **(a)** and $10^\circ$ S–$10^\circ$ N **(d)**, the extratropics $30$–$60^\circ$ N/S **(b)/(e)** and the polar region $60$–$90^\circ$ N/S **(c)/(f)**.



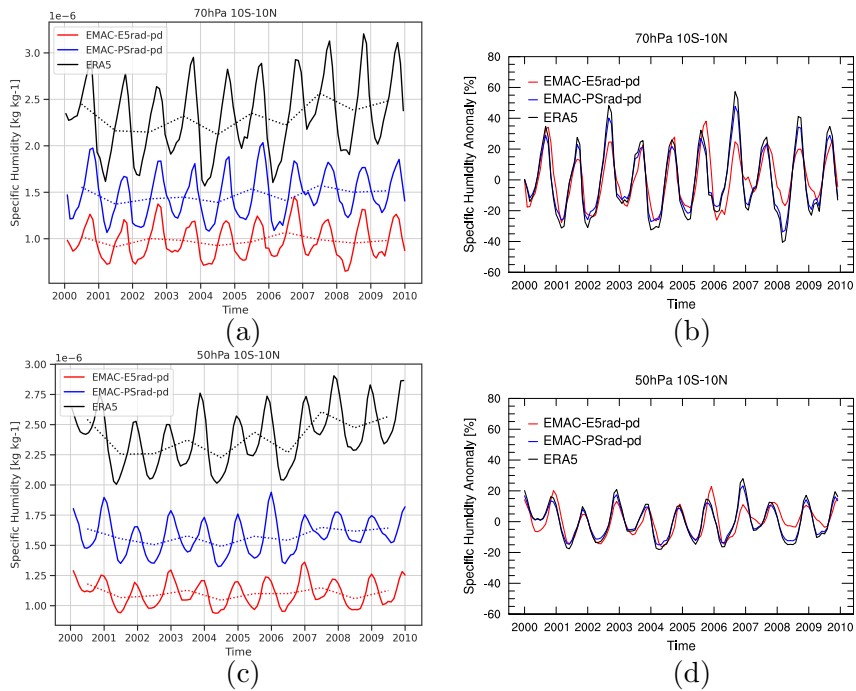

**Figure 8.** Tape recorder signal at $70\,\mathrm{hPa}$ (top row) and $50\,\mathrm{hPa}$ (bottom row) given by the specific humidity averaged over $10°\,\mathrm{S}$–$10°\,\mathrm{N}$. **(a, c)** Time series of specific humidity in $10^{-6}\,\mathrm{kg\,kg^{-1}}$. **(b, d)** Relative anomaly (in percent) of the tape recorder signal, i.e. displayed is the relative anomaly with respect to the respective long-term mean (all months weighted equally).

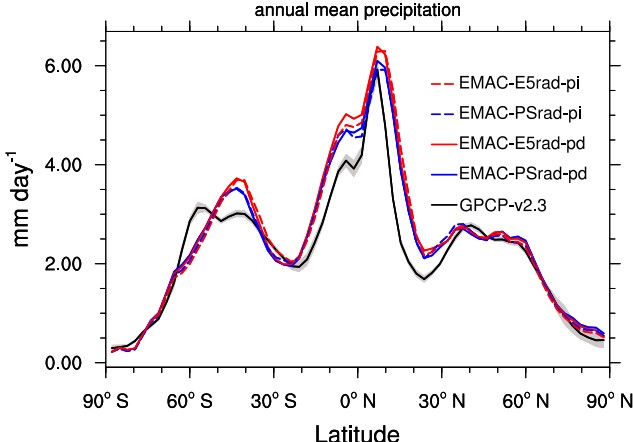

**Figure 9.** Multiannual zonal mean precipitation $(\mathrm{mm\,day^{-1}})$ for the last ten years of the simulations and the period 2000–20009 for GPCP v2.3 data. GPCP v2.3 was conservatively regridded to a T42 grid using Climate Data Operators (CDO, https://code.mpimet.mpg.de/projects/cdo/ last accessed 21 August 2023).





## 4 Radiative forcing calculations using multiple diagnostic calls

We use the simulations of the newly optimized model configurations to assess RFs due to perturbations of GHGs in the old and new model setups.[1] A central objective of the intercomparison presented here is to enable the attribution of differing RF results either to differences in the background meteorology or to differences in the actual radiative transfer calculation, as well as to assess the impact of different GHG backgrounds on the RF values related to a perturbation. To this end, additional diagnostic calls of the radiation scheme with perturbed GHGs (namely, $CO_2$, $N_2O$, $CH_4$ and CFCs) have been conducted in both the simulations under pre-industrial and present-day conditions, which employ once the ECHAM5 radiation scheme (E5rad) and once the PSrad radiation scheme for driving the simulation. The respective GHG fields were adopted from previous EMAC simulations (see Tab. 2), except for the methane field which enters the first call of the radiative transfer calculation and drives the simulation (see Sect. 3.1).

Table 5 lists the respective perturbations that are calculated in the multiple calls of the radiation scheme. In total, 22 additional (diagnostic) calls for calculating instantaneous RF (calls 02 to 23) and 11 additional calls for calculating stratospheric adjusted RF (calls 24 through 34) have been conducted. In the columns of Table 5 the perturbations are listed, for example for call 03 (call 25), $CO_2$ has been set to present-day values for the pi simulations and to pre-industrial values for the pd simulations. Thus, the instantaneous (stratospheric adjusted) RF due to increasing $CO_2$ from pre-industrial to present-day levels can be assessed from $F_{pi;CO_2(pd)} - F_{pi}$ or alternatively from $F_{pd} - F_{pd;CO_2(pi)}$. The first subscript denotes the reference state, the second subscript (if present) denotes the species that has been perturbed and $F$ denotes the instantaneous (stratospheric adjusted) TOA radiative fluxes from call 02 and call 03 (call 24 and call 25), respectively. This may be viewed as the adoption of the forward and backward calculation method (known from radiative feedback analysis, for example, Colman and McAvaney, 1997; Klocke et al., 2013; Rieger et al., 2017) for the RF calculation, which allows to assess the effect of the GHG background on the diagnosed forcing.

Additionally, for the calculation of instantaneous RFs diagnostic calls with a "switched" radiation scheme have been performed. This means that the radiation scheme driving the simulation and the radiation scheme used in a diagnostic call are different. For example, calls 13 and 14 from the EMAC-E5rad-pi simulation can be used to evaluate the instantaneous RF of present-day $CO_2$ using the PSrad radiation scheme in a pre-industrial simulation, which is driven by E5rad. This provides the opportunity to further assess the dependence of the RF results on the background (here this does not refer to present-day vs. pre-industrial but rather the different meteorological climatologies from the models that serve as different backgrounds) or the

---

[1]We denote all flux changes resulting from perturbations of GHGs with RF, although RF is often recommended for use with respect to the pre-industrial reference state, especially within the CMIP framework (Pincus et al., 2016), in order to ensure optimal comparability in multi-model intercomparison studies as e.g. by Ceppi et al. (2017) and Zelinka et al. (2020). We follow the less strict definition of Fuglestvedt et al. (e.g 2010) and Ramaswamy et al. (2018), according to which the use of any quasi-stationary reference state is appropriate. This notion emphasizes the role of RF as a predictor of expected global mean equilibrium surface temperature change (e.g. Hansen et al., 2005).





employed radiation scheme.

**Table 5.** Employed radiation perturbations for the four EMAC simulations. The first call drives the respective simulation, calls 02–12 are used for calculating various instantaneous RFs due to the perturbation of GHGs. Calls 13–23 allow to assess the RFs of the same perturbation with the switched radiation scheme, whereas calls 24–34 allow to assess the stratospheric adjusted RFs.

| Call | EMAC-E5rad-pi | EMAC-E5rad-pd | EMAC-PSrad-pi | EMAC-PSrad-pd |
|---|---|---|---|---|
| 01 | base | base | base | base |
| 02-23 | instantaneous | | | |
| 02/13* | base | base | base | base |
| 03/14 | $CO_2$(pd) | $CO_2$-pi | $CO_2$(pd) | $CO_2$-pi |
| 04/15 | $N_2O$(pd) | $N_2O$(pi) | $N_2O$(pd) | $N_2O$(pi) |
| 05/16 | $CH_4$(pd) | $CH_4$(pi) | $CH_4$(pd) | $CH_4$(pi) |
| 06/17 | CFC(pd) | CFC(pi) | CFC(pd) | CFC(pi) |
| 07/18 | 2x$CO_2$(pi) | 2x$CO_2$(pd) | 2x$CO_2$(pi) | 2x$CO_2$(pd) |
| 08/19 | 4x$CO_2$(pi) | 4x$CO_2$(pd) | 4x$CO_2$(pi) | 4x$CO_2$(pd) |
| 09/20 | 2x$CH_4$(pi) | 2x$CH_4$(pd) | 2x$CH_4$(pi) | 2x$CH_4$(pd) |
| 10/21 | 5x$CH_4$(pi) | 5x$CH_4$(pd) | 5x$CH_4$(pi) | 5x$CH_4$(pd) |
| 11/22 | 2x$N_2O$(pi) | 2x$N_2O$(pd) | 2x$N_2O$(pi) | 2x$N_2O$(pd) |
| 12/23 | 5x$N_2O$(pi) | 5x$N_2O$(pd) | 5x$N_2O$(pi) | 5x$N_2O$(pd) |
| 24-34 | stratospheric adjusted | | | |
| 24 | base | base | base | base |
| 25 | $CO_2$(pd) | $CO_2$(pi) | $CO_2$(pd) | $CO_2$(pi) |
| 26 | $N_2O$(pd) | $N_2O$(pi) | $N_2O$(pd) | $N_2O$(pi) |
| 27 | $CH_4$(pd) | $CH_4$(pi) | $CH_4$(pd) | $CH_4$(pi) |
| 28 | CFC(pd) | CFC(pi) | CFC(pd) | CFC(pi) |
| 29 | 2x$CO_2$(pi) | 2x$CO_2$(pd) | 2x$CO_2$(pi) | 2x$CO_2$(pd) |
| 30 | 4x$CO_2$(pi) | 4x$CO_2$(pd) | 4x$CO_2$(pi) | 4x$CO_2$(pd) |
| 31 | 2x$CH_4$(pi) | 2x$CH_4$(pd) | 2x$CH_4$(pi) | 2x$CH_4$(pd) |
| 32 | 5x$CH_4$(pi) | 5x$CH_4$(pd) | 5x$CH_4$(pi) | 5x$CH_4$(pd) |
| 33 | 2x$N_2O$(pi) | 2x$N_2O$(pd) | 2x$N_2O$(pi) | 2x$N_2O$(pd) |
| 34 | 5x$N_2O$(pi) | 5x$N_2O$(pd) | 5x$N_2O$(pi) | 5x$N_2O$(pd) |

∗ first number refers to the call with the driving radiation scheme, second number to the call with the switched radiation scheme.

Table 6 shows the instantaneous and stratospheric adjusted RF means for the last 10 years of the simulation for different

GHG perturbations. In the calls in which a single GHG is doubled, quadrupled or quintupled, the increase relates to the respective base period of the simulations, i.e. for the 2x$CH_4$ experiments the $CH_4$(pi) values have been doubled for the pi simulations,



whereas the $CH_4$(pd) values have been doubled for the pd simulations. Note that in this table the forcings are calculated with the same radiation scheme that is also driving the dynamic simulation. For instantaneous RFs, we will also address (somewhat below) the results from RF calculations, which result from switching the radiation scheme (Tab. 8).


We start our evaluation by comparing stratospheric adjusted RFs from our simulations (columns 2 to 5 in Table 6) with idealized estimates (two rightmost columns in Table 6), which are based on formulas presented by Etminan et al. (2016). Overall the results from the simulations using PSrad are closer to the Etminan-based estimates concerning stratospheric adjusted RF. In particular, this is true for the assessment of stratospheric adjusted RFs from $CH_4$(pi) and 2x$CH_4$, which are substantially

higher in PSrad than in E5rad, and for 4x$CO_2$, which are lower in PSrad than in E5rad. We note here that the estimates given in brackets are outside the recommended range of the formulas as indicated by Etminan et al. (2016). We nevertheless present these values as they provide additional evidence that the PSrad scheme yields much more realistic stratospheric adjusted RF values, especially for $CH_4$ and (see below) $N_2O$ perturbations. The instantaneous and stratospheric adjusted RF values due to doubling or quadrupling $CO_2$ from the EMAC-E5rad-pd simulation are in agreement with previous results obtained with

EMAC and the ECHAM5 radiation scheme as presented by Dietmüller et al. (2014) and Rieger et al. (2017; see the forward results in both studies).

Additional stratospheric adjusted and instantaneous RFs for 2x$CO_2$ and 3x$CH_4$ from global model simulations have been presented by Richardson et al. (2019). Please see their Section 2 on how the respective forcings were defined and note that

they (mostly but not exclusively) use present-day as the reference state. For the latter reason, we will address results from our pd simulations for comparisons only. For the 2x$CO_2$ RFs, the results from our EMAC-PSrad-pd simulation are closer to the values presented by Richardson et al. (2019) than the RFs based on EMAC-E5rad-pd for both instantaneous and stratospheric adjusted RFs. For 3x$CH_4$ RFs the results from our EMAC-E5rad-pd simulation ($0.24\,\mathrm{W\,m^{-2}}$ and $0.3\,\mathrm{W\,m^{-2}}$ for instantaneous RF and stratospheric adjusted RF, respectively; interpolated from the 2x$CH_4$ and 5x$CH_4$ RFs) show clearly lower values than

the results from the EMAC-PSrad-pd simulation ($0.97\,\mathrm{W\,m^{-2}}$ and $0.95\,\mathrm{W\,m^{-2}}$, respectively; interpolated as before). The increased RFs associated with a 3x$CH_4$ experiment as diagnosed from PSrad are in better agreement with the values presented by Richardson et al. (2019), which are somewhat above $1\,\mathrm{W\,m^{-2}}$.

Another aspect to note about the methane RFs is that with PSrad the stratospheric temperature adjustment acts to reduce

the RF in comparison with the instantaneous RF, whereas for E5rad it acts to increase it. PSrad includes SW absorption of methane in two bands in the near-infrared (3.08 - 3.85 $\mu$m and 2.15 - 2.50 $\mu$m; cf. the RRTM bands described in the ECHAM6 documentation Giorgetta et al., 2013b). The SW absorption acts to counteract the stratospheric cooling induced by the LW radiation (Byrom and Shine, 2022, their Fig. 2). Similarly, Smith et al. (2018, their Fig. S6) found that for the same experiments as analysed by Richardson et al. (2019), the rapid radiative adjustment induced by the stratospheric temperature adjustment

is negative in models with the explicit treatment of methane SW absorption in the radiation scheme, and positive in models



without.

The instantaneous RF of $3xN_2O$ with respect to present-day conditions has been assessed by Hodnebrog et al. (2020) for global models and LBL calculations. They find an instantaneous RF of roughly $1.5\,\mathrm{W\,m^{-2}}$ and $1.4\,\mathrm{W\,m^{-2}}$, respectively. In-
terpolation of the instantaneous $2xN_2O$ and $5xN_2O$ calculations from EMAC-E5rad-pd and EMAC-PSrad-pd yields values of $2.49\,\mathrm{W\,m^{-2}}$ and $1.37\,\mathrm{W\,m^{-2}}$, respectively, clearly emphasizing the superiority of $N_2O$ forcings provided by the latter.

**Table 6.** RFs $(\mathrm{W\,m^{-2}})$ for perturbations based on the diagnostic radiation calls described in Table 5 for the last 10 years of the simulations. In addition best estimates based on the formula from Etminan et al. (2016) are given as reference values for stratospheric adjusted RF.

| | EMAC-E5rad-pi | EMAC-E5rad-pd | EMAC-PSrad-pi | EMAC-PSrad-pd | Etminan pi | Etminan pd |
|---|---|---|---|---|---|---|
| Perturbation | instantaneous RF $(\mathrm{W\,m^{-2}})$ | | | | | |
| $CO_2(pi)$ | 0.86 | 0.94 | 0.75 | 0.81 | | |
| $N_2O(pi)$ | 0.22 | 0.21 | 0.19 | 0.16 | | |
| $CH_4(pi)$ | 0.24 | 0.25 | 0.41 | 0.39 | | |
| $CFC(pi)$ | 0.24 | 0.25 | 0.29 | 0.29 | | |
| $2xCO_2$ | 2.34 | 2.65 | 1.93 | 2.13 | | |
| $4xCO_2$ | 5.04 | 5.77 | 3.85 | 4.24 | | |
| $2xCH_4$ | 0.21 | 0.16 | 0.35 | 0.58 | | |
| $5xCH_4$ | 0.42 | 0.39 | 1.15 | 1.75 | | |
| $2xN_2O$ | 1.34 | 1.41 | 1.03 | 0.87 | | |
| $5xN_2O$ | 4.44 | 4.65 | 2.64 | 2.37 | | |
| Perturbation | stratospheric adjusted RF $(\mathrm{W\,m^{-2}})$ | | | | | |
| $CO_2(pi)$ | 1.44 | 1.45 | 1.38 | 1.39 | 1.40 | 1.39 |
| $N_2O(pi)$ | 0.23 | 0.21 | 0.20 | 0.17 | 0.14 | 0.13 |
| $CH_4(pi)$ | 0.29 | 0.29 | 0.42 | 0.38 | 0.53 | 0.53 |
| $CFC(pi)$ | 0.23 | 0.25 | 0.27 | 0.27 | - | - |
| $2xCO_2$ | 4.02 | 4.23 | 3.80 | 3.91 | 3.80 | 3.83 |
| $4xCO_2$ | 8.61 | 9.12 | 7.88 | 8.07 | 7.96 | 8.04 |
| $2xCH_4$ | 0.26 | 0.20 | 0.36 | 0.57 | 0.46 | (0.64) |
| $5xCH_4$ | 0.54 | 0.50 | 1.16 | 1.70 | (1.32) | (1.74) |
| $2xN_2O$ | 1.38 | 1.45 | 1.08 | 0.92 | (0.77) | (0.79) |
| $5xN_2O$ | 4.62 | 4.83 | 2.78 | 2.50 | (2.33) | (2.40) |

The interannual standard deviations were in the order of $0.01\,\mathrm{W\,m^{-2}}$. Values in brackets in the columns Etminan-pi and Etminan-pd are for perturbations that are outside the valid range of the approximation formulas given by Etminan et al. (2016). The perturbations $2xN_2O(pi)$ and $2xCH_4(pd)$ are close to the valid range.



Table 7 shows the global mean clear-sky instantaneous RFs corresponding to the all-sky instantaneous RFs presented in Table 6. Our results can be compared with those from Pincus et al. (2020), which were derived from the multi-model mean
of so-called "benchmark" models. Based on the description by Pincus et al. (2020), we can compare the results from EMAC-E5rad-pd and EMAC-PSrad-pd shown in Table 7 with their results for clear-sky instantaneous RF due to increasing a single GHG from pre-industrial to present-day values. However, as the base periods and values for pi and pd conditions are different, for example, Pincus et al. (2020) use 2014 as pd, we rescaled our clear-sky RF results to allow for a better comparison. The corresponding values are listed in brackets in Table 7. For the rescaling, we assumed that the 2014 values used by Pincus et al.
(2020) are similar to the values presented by Meinshausen et al. (2017). Consequently, the clear-sky instantaneous RFs were adjusted as follows: $iRF_{cs}^* = iRF_{cs} \cdot \Delta X_{P20}/\Delta X_{N23}$, where $iRF_{cs}$ refers to the instantaneous clear-sky RF and the asterisk denotes the corresponding rescaled quantity. $\Delta X$ denotes the change (in $\mathrm{mol\,mol}^{-1}$) of the species $X$ from pi to pd conditions and the subscripts P20 and N23 refer to Pincus et al. (2020) and our study, respectively. Taking into account the rescaling, all clear-sky RFs for the pi experiments calculated with PSrad are closer to the results presented by Pincus et al. (2020) than the
results obtained with E5rad. As an example, the global mean clear-sky RF (including the above-mentioned correction) due to the rise of methane from pi to pd increases from $0.41\,\mathrm{W\,m}^{-2}$ in the E5rad simulation to $0.51\,\mathrm{W\,m}^{-2}$ in the simulation with PSrad and is closer to the reference value of $0.67\,\mathrm{W\,m}^{-2}$ presented by Pincus et al. (2020). Conversely, for N$_2$O the clear-sky instantaneous RF decreases when PSrad is used and is thus in better agreement with the value presented by Pincus et al. (2020).

Pincus et al. (2020) also show clear-sky RFs with respect to CO$_2$-folding experiments. Presuming that they use pre-industrial CO$_2$ as a reference state for CO$_2$, whereas the other GHGs and the meteorology are representative of present-day conditions, one can try to compare their results with our rescaled results for the CO$_2$-folding experiments performed in the EMAC-E5rad-pi and EMAC-PSrad-pi simulations. This would lead to a seemingly better agreement of E5rad than PSrad results with their values. However, we warrant that this comparison is questionable due to the following: (i) We have a different GHG (including
water vapour) background, namely pi, in comparison to their background of pd conditions. We assume that through compensation we would get lower RFs (i.e. less sensitivity to CO$_2$ changes) than presented here, if the CO$_2$-folding would have been performed against a pd GHG background. (ii) In the climatological pd background, the temperatures are likely higher than our pi background. Here, we reason that this will likely lead to an increased RF as diagnosed from CO$_2$-folding experiments, as the surface emits more LW radiation which can be attenuated by the additional CO$_2$.


An estimate of the combined effect can be obtained when comparing our "forward" and "backward" experiments for calculating the clear-sky RF due to the increase of a single GHG from pi to pd levels. For both, E5rad and PSrad, the clear-sky RF due to the rise of CO$_2$ from pi to pd levels is higher, when assessed against the pd background. For N$_2$O the relation is reversed, whereas for CFCs there is (almost) no dependence of the instantaneous clear-sky RF on the background. Interestingly, for CH$_4$
the clear-sky instantaneous RFs are higher for a pd background when assessed with E5rad, and lower when assessed with PSrad compared to the RFs when calculated against a pi background. Qualitatively similar dependencies of the instantaneous



RFs on the GHG background are found for the all-sky fluxes (see Table 6).

**Table 7.** Global mean clear-sky instantaneous RFs ($\mathrm{W\,m^{-2}}$) for perturbations based on the diagnostic radiation calls described in Table 5 for the last 10 years of the simulations. Values in brackets denote rescaled EMAC clear-sky RFs, which are supposed to ensure better comparability with the RFs presented by Pincus et al. (2020). See text for details.

| | EMAC-E5rad-pi | EMAC-E5rad-pd | EMAC-PSrad-pi | EMAC-PSrad-pd |
|---|---|---|---|---|
| Perturbation | clear-sky instantaneous RF ($\mathrm{W\,m^{-2}}$) | | | |
| $CO_2$(pi) | 1.04 | 1.11 (1.57) | 0.97 | 1.03 (1.46) |
| $N_2O$(pi) | 0.27 | 0.25 (0.32) | 0.24 | 0.19 (0.24) |
| $CH_4$(pi) | 0.33 | 0.35 (0.41) | 0.48 | 0.44 (0.51) |
| CFC(pi) | 0.33 | 0.34 (0.43) | 0.38 | 0.38 (0.48) |
| $2xCO_2$ | 2.81 | 3.12 | 2.55 | 2.76 |
| $4xCO_2$ | 5.96 | 6.65 | 5.17 | 5.56 |
| $2xCH_4$ | 0.30 | 0.23 | 0.41 | 0.67 |
| $5xCH_4$ | 0.60 | 0.54 | 1.34 | 2.01 |
| $2xN_2O$ | 1.63 | 1.70 | 1.27 | 1.07 |
| $5xN_2O$ | 5.37 | 5.59 | 3.27 | 2.93 |

The interannual standard deviations were in the order of 0.01 $\mathrm{W\,m^{-2}}$.

The instantaneous RFs presented in Table 6 are complemented by Table 8, which arises when the instantaneous RF is calcu-
lated with a different radiation scheme compared to the scheme that is driving the simulation. Hence, the columns of Table 6 and Table 8 can be compared to each other one to one. Overall the relative differences are roughly 10% or less, showing that the results are relatively robust to changes in the background state related to switching the radiation scheme. With respect to experiments, that assess the instantaneous RF due to an increase of a single GHG from pi to pd levels, we find that the changes of the meteorological background associated with the radiation scheme do not play a major role: For CFCs, $N_2O$ and $CH_4$ they are almost negligible whereas they are somewhat larger for $CO_2$ (the respective values in Table 6 and Table 8 are almost identical except for the $CO_2$ perturbations).





**Table 8.** Instantaneous RFs for perturbations described in Table 5 for the last 10 years of the simulations, where the radiation scheme for diagnosing the instantaneous RF was switched compared to the radiation scheme driving the simulation. As an example: in the second column radiation calls with the E5rad scheme were used to calculate the instantaneous RFs within the EMAC-PSrad-pi simulation.

| Simulation | EMAC-PSrad-pi | EMAC-PSrad-pd | EMAC-E5rad-pi | EMAC-E5rad-pd |
|---|---|---|---|---|
| Radiation scheme | E5rad | E5rad | PSrad | PSrad |
| Perturbation | instantaneous RF ($W\,m^{-2}$) | | | |
| $CO_2$(pi) | 0.81 | 0.89 | 0.80 | 0.86 |
| $N_2O$(pi) | 0.22 | 0.21 | 0.20 | 0.16 |
| $CH_4$(pi) | 0.24 | 0.26 | 0.42 | 0.39 |
| CFC(pi) | 0.24 | 0.25 | 0.29 | 0.29 |
| $2xCO_2$ | 2.19 | 2.50 | 2.08 | 2.28 |
| $4xCO_2$ | 4.72 | 5.45 | 4.18 | 4.58 |
| $2xCH_4$ | 0.22 | 0.16 | 0.36 | 0.59 |
| $5xCH_4$ | 0.42 | 0.38 | 1.18 | 1.78 |
| $2xN_2O$ | 1.32 | 1.38 | 1.04 | 0.88 |
| $5xN_2O$ | 4.35 | 4.56 | 2.67 | 2.40 |

The interannual standard deviations were in the order of 0.01 $W\,m^{-2}$



# 5 Summary and conclusions

In this paper, we describe the recent upgrades of the MESSy radiation infrastructure and its first applications. In Sect. 2 we give a detailed overview of the implemented changes. A guideline through the implementation process has been to retain the possibility to use all previous model setups (backward compatibility) and to ensure the applicability of MESSy-specific features (e.g. multiple radiation calls) also with the updated radiation infrastructure. Specific highlights of the new implementations are the integration of the radiation scheme PSrad and the availability of a new submodel ALBEDO, which features solar zenith angle dependent albedos. Further, a white-sky albedo for snow-free land has been compiled based on satellite data.

The third Section (Sect. 3.2) exemplarily describes the model optimization of a typical "old" (with ECHAM5 radiation) and "new" (with PSrad radiation) dynamical model setup (fixed sea surface temperatures and no chemistry except for simplified methane chemistry) with a consistent set of parameters for pre-industrial and present-day conditions. Comparing the old and new setup, also with observational and reanalysis data, shows that the main features of the simulated climate (also known from previous ECHAM5 and other ECHAM6.1 simulations, e.g. Stevens et al., 2013) remain. However, some biases of the old model setup, e.g. the cold bias in the tropical upper troposphere–lower stratosphere and a too weak polar vortex, seem to be reduced when the PSrad scheme is employed.

Finally, we show radiative forcing results based on the old and the new model setups using multiple diagnostic radiation calls. In total we perform 33 additional diagnostic radiation calls per simulation to assess various radiative forcings. In particular, we show stratospheric (temperature) adjusted and instantaneous RF values due to reduced or increased greenhouse gases. When comparing these results with previous estimates, we find that PSrad generally performs better for instantaneous and stratospheric-adjusted radiative forcings. In particular, methane (nitrous oxide) radiative forcings calculated with PSrad are much increased (decreased) to the radiative forcings calculated with the ECHAM5 radiation scheme, which means a clear improvement when compared to benchmark results. For the instantaneous forcings we also derive results where the radiation scheme of the diagnostic calls is switched compared to the driving radiation scheme, i.e. using the old radiation scheme to propagate the simulation and evaluating two additional diagnostic radiation calls with the new radiation scheme to determine the instantaneous flux changes or vice versa. It appears that changes in the radiative forcing results from the previous (ECHAM5) setup to the new (PSrad) setup are mainly attributable to the radiative transfer calculations themselves, whereas the changed background climatology related to the driving radiation scheme plays only a minor role.

The implemented changes lead to an improved representation of tropical upper tropospheric temperatures (and thus stratospheric water vapour). Further, various radiative forcings due to greenhouse gas perturbations tend to be improved. In particular, this is the case for methane forcing experiments, which show a higher radiative forcing with the new radiation scheme, PSrad, and are thus in better agreement with literature-based reference values. The latter can be exploited to better quantify methane radiative forcings and the role of methane as a feedback component in the climate system. The developments mark an impor-



tant step for the MESSy framework to be able to include additional radiation schemes. The next steps concerning the use of the MESSy radiation infrastructure are to employ the PSrad scheme with interactive chemistry and an online coupled ocean (Earth system model setup). Further envisaged developments are the coupling of PSrad to FUBrad and the use of PSrad with

an interactive aerosol model, which will be enabled by the revision of the AEROPT submodel.

*Code availability.* The Modular Earth Submodel System (MESSy) is continuously further developed and applied by a consortium of institutions. The usage of MESSy and access to the source code is licenced to all affiliates of institutions which are members of the MESSy Consortium. Institutions can become a member of the MESSy Consortium by signing the MESSy Memorandum of Understanding. More

information can be found on the MESSy Consortium Website (http://www.messy-interface.org). The code presented here has been based on MESSy version 2.55 and will be available in the next official release (version 2.56).

*Data availability.* GPCP v2.3 data were downloaded from https://psl.noaa.gov/data/gridded/data.gpcp.html (downloaded 15 February 2023, last access 17 February 2023). MODIS MCD43GF v006 data (MODIS/Terra+Aqua BRDF/Albedo Gap-Filled Snow-Free Daily L3 Global 30ArcSec CMG) from the NASA EOSDIS Land Processes Distributed Active Archive Center (LP DAAC; Schaaf, 2019) located at the

USGS Earth Resources Observation and Science (EROS) Center have been obtained from the Data Pool (https://e4ftl01.cr.usgs.gov/MOTA/ MCD43GF.006). HadISST data were obtained from https://www.metoffice.gov.uk/hadobs/hadisst/ and are © British Crown Copyright, Met Office, [2023], provided under a Non-Commercial Government Licence http://www.nationalarchives.gov.uk/doc/non-commercial-government-licence/ version/2/. Contains modified Copernicus Climate Change Service information [2023]. Neither the European Commission nor ECMWF is responsible for any use that may be made of the Copernicus information or data it contains. ERA5 monthly mean data on pressure levels

(Hersbach et al., 2023) were downloaded from the Copernicus Climate Change Service (C3S) Climate Data Store (Copernicus Climate Change Service, Climate Data Store, 2023). Various CMIP6 data (e.g. Meinshausen and Vogel, 2016; Matthes et al., 2017b; ETH Zürich (ETHZ), 2017) used as boundary conditions (also for previous EMAC simulations) are available from ESGF.

*Author contributions.* MN implemented RAD/CLOUDOPT with help of PJ and LS. LS and PJ implemented the ALBEDO submodel with the help of FW and MN. LS adjusted AEROPT with the help of PJ and MN. Preparation of model setups for parameter optimization and

evaluation: LS, MN, PJ, PG, MK and FW. Conduction of simulations: LS. Data analysis: LS, FW and MN. MN drafted and wrote main parts of the paper with help of LS, FW and MP. All authors contributed to the discussion of the results and/or to the developments described in the paper.

*Competing interests.* At least one of the (co-)authors is a member of the editorial board of Geoscientific Model Development. The authors have no other competing interests to declare.



*Acknowledgements.* We are thankful for helpful clarifications from Sebastian Rast (MPI-M) regarding various ICON and ECHAM related topics. We thank Ralf Meerkötter (DLR) for pointing out the importance of the SZA dependence of the surface albedos, Birgit Hassler (DLR) for the internal review, Mattia Righi (DLR) for assistance and helpful comments regarding the optimization procedure, and the EVA department at DLR for help with the ESMValTool. We thank Simone Dietmüller (DLR) for helpful discussions on stratospheric adjusted radiative forcing and Bernhard Mayer (LMU) for helpful discussions e.g. on methane radiative forcing. The analysis of simulation results

was supported by the ESMValTool 2.8 (doi:10.5281/zenodo.3401363) and ESMValCore 2.8 (doi:10.5281/zenodo.3387139) (Righi et al., 2020). We used Climate Data Operators (CDO; https://code.mpimet.mpg.de/projects/cdo/, last access: 21 August 2023; Schulzweida, 2022) for data processing and the NCAR Command Language (NCL, 2019, see references) for parts of the data analysis. Global Precipitation Climatology Project (GPCP) Monthly Analysis Product data were provided by the NOAA PSL, Boulder, Colorado, USA, from their website at https://psl.noaa.gov (last access 17 February 2023). We thank the NASA EOSDIS Land Processes Distributed Active Archive Center

(LP DAAC) for making MODIS MCD43GFv006 data available (Schaaf, 2019). HadISST data (Rayner et al., 2003) are available from www.metoffice.gov.uk/hadobs (last access 06 Mar 2023). We acknowledge the World Climate Research Programme, which, through its Working Group on Coupled Modelling, coordinated and promoted CMIP6. We thank the climate modeling groups for producing and making available their model output, the Earth System Grid Federation (ESGF) for archiving the data and providing access, and the multiple funding agencies who support CMIP6 and ESGF. The simulations have been performed on the DLR HPC-cluster CARA.

*Financial support.* The work described in this paper has received funding from the Initiative and Networking Fund of the Helmholtz Association through the project "Advanced Earth System Modelling Capacity (ESM)" and from the Helmholtz Association project "Joint Lab Exascale Earth System Modelling (JL-ExaESM)". The content of the paper is the sole responsibility of the author(s) and it does not represent the opinion of the Helmholtz Association, and the Helmholtz Association is not responsible for any use that might be made of the information contained. Further funding was received from the DFG through the project IRFAM-ClimS (Vorhaben WI 5369/1-1). PG acknowledges

funding from the German Federal Ministry of Education and Research (BMBF) as part of the "Research for Sustainability (FONA)" strategy for the project entitled "The Climate Model Intercomparison Project 6 - Chemistry (CMIP6-Chemistry, Förderkennzeichen 01LP1606A).



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
