# Peer review of "Updating the radiation infrastructure in MESSy (based on MESSy version 2.55)"

_EGUsphere, 2023_

## Author Comment (AC1)

**Author Comment to manuscript egusphere-2023-2140, (https://doi.org/10.5194/egusphere-2023-2140, in review, 2023): "Updating the radiation infrastructure in MESSy (based on MESSy version 2.55)"**

by M. Nützel et al.

February 9, 2024

We thank the referees for taking the time to review our paper. We are grateful for their comments which helped to improve the manuscript. In the following we address each review comment (*black italics*) by stating our reply (blue). In addition we append a manuscript version which highlights the changes between the preprint version of the manuscript and the revised version.

**Reply to comments from editor**

In addition to the comments by the reviewers, the editor has commented on our discussion version and requested these comments to be considered in a revised version. We thank the editor for these comments which we will address below.

*Minor comments on egusphere-2023-2140*

*1. Abstract: The statement "they also aim towards the use of MESSy with the ICOsahedral Non- hydrostatic (ICON) model" is unclear. I think it means that the use of this development will be feasible in the MESSy infrastructure,*

*using ICON as the base model, but it should be more clearly written.*

We rephrased the sentence which now reads: "The developments presented here also aim towards the use of the MESSy infrastructure with the ICOsahedral Non-hydrostatic (ICON) model as a base model." We hope that this removes any ambiguities.

*2. Line 27: Correct spelling of "asessed"*

Done.

*3. Line 55: No need for "radiative" in front of "RFs"*

Done.

*4. Line 190: I would recommend changing "supposed to follow". Often, the word "supposed" can have a negative context, i.e., something was planned, but it didn't actually happen! How about "this functionality is due to be implemented with a revision of the AEROPT submodel"?*

Done.

*5. Line 255: Change "Still missing" to "Any remaining missing"*

Done.

*6. Line 278/298: Change "via namelist" to "via a namelist"*

Done.

*7. Line 296: Change "where shifted" to "were shifted"*

Done.

*8. Line 469: Please provide full name for JJA on first use (Same applies for DJF on line 476)*

61    Done.

63    *9. Line 523: Please correct the bracketing*

65    We could not find any bracketing that needs correction in line 523. The
66    formula for calculating relative anomalies is correct and we assume that the
67    bracket "(panels b and d)" is also ok.

69    *10. Line 660: I suggest that you replace "guideline" with "guiding principle".*

71    Done.

72

**Reply to comments from CEE (https://doi.org/10.5194/egusphere-2023-2140-CEC1)**

The executive editor has commented on our discussion version. We will address this comment below.

*Dear authors,*

*Please, in any potential reviewed version of your manuscript provide in the "Code Availability" section a link to the MESSY private repository in Zenodo, including its DOI.*

*Best regards,*

*Juan A. Añel*

*Geosci. Model Dev. Executive Editor*

We thank the executive editor for this comment. The respective reference is now included in the "Code Availability" Section. Please note that these updates are not highlighted in the appended diff-version.

**Reply to comments from Referee #1 (https://doi.org/10.5194/egusphere-2023-2140-RC1)**

Below we will address all comments of referee #1 and we will state corresponding changes in the manuscript. Again, we would like to thank referee #1 for taking the time to review our manuscript and for the thoughtful comments.

*This paper is, in part, a technical report of the updated infrastructure concerning the treatment of radiation in the Modular Earth Submodel System (MESSy), and in part, an evaluation of the performance of the newly implemented PSrad (Pincus and Stevens) radiation scheme vs. the ECHAM5 radiation scheme. It is clearly written with sufficient technical detail to be useful for developers of the MESSy infrastructure as well as serving as a useful example for developers of other model radiation schemes.*

*The evaluation of the radiation schemes serves as a good test of the implementation and a useful evaluation of two schemes side-by-side in an identical model. The only problematic area is the comparison of the schemes against reference data presented in Pincus et al (2020), based on RFMIP (Radiative Forcing Model Intercomparison Project).*

*I would recommend this paper for publication once the following, generally minor comments have been addressed:*

We thank the reviewer for this general rating of our manuscript. We revised our document according to the suggestions given by the reviewer and here we reply to each of the comments. In particular we tried to adjust the comparison to reference data. If we did not follow the suggestions at some particular instance we hope that our respective replies make our choice understandable.

*Principal comment:*

*1) Section 4, lines 630-640: The arguments presented here may be valid but it feels like the overall argument in this section is biased towards achieving a better comparison for the PSrad scheme. I think a more robust comparison could be done avoiding the need for the caveats in this section.*

*In the previous paragraph, lines 613-628, you use your present-day (PD)*
*background runs to compare with the Pincus et al results for the forcing from*
*pre-industrial to present-day GHG amounts. You scale the quantities to account*
*for the different PD background conditions which sounds reasonable. For the*
*CO2-folding experiments, however, you revert to the pre-industrial (PI) back-*
*ground runs. Your following arguments detail why this is a bad thing to do.*
*Given that you have a range of CO2-folding experiments for the PD-background*
*runs: CO2(pi), CO2(pd), 2xCO2(pd), 4xCO2(pd), you should be able to inter-*
*polate values for 2xCO2(pi) and 4xCO2(pi) to directly compare with Pincus et*
*al. It would then be good to have all the Pincus et al results listed in table 7 to*
*provide a clear comparison for the reader.*

We thank the reviewer for this comment and in particular for bringing the
option of using our pd background simulation for comparison to our atten-
tion. We think that scaling the pi-pd results is reasonable, which can be seen
as a comparison of radiative efficiencies. With respect to the $CO_2$-folding ex-
periments, we referred to the pi simulation in which we did $CO_2$(pi)-folding
experiments because this minimizes the differences between the sampling points
with respect to which RF is calculated. We thought that this might be the
first point to go to when comparing our study to the results by Pincus et al.
(2020). Hence we warrant, that this of course comes at the drawback of having
a different background. Doing the analysis the other way round - as suggested
by the reviewer - leads to a comparable (pd) background at the expense of hav-
ing the $CO_2$-folding experiments at sampling points which are quite different
from the ones used in Pincus et al. (2020). As we were driven by comparing
at similar sampling points, we completely disregarded the option raised by the
reviewer. We now added a figure to our paper and discuss this second option.
Nevertheless, we will also keep the discussion of the first option as we think it
is good to put this approach into perspective and to outline the possible caveats.

*Minor comments:*

*1) Section 1, line 89: "resulted in 0.23 Wm-2": please define what this num-*
*ber represents, i.e. define radiative forcing as the difference in which fluxes?*
*Top-of-atmosphere / tropopause / surface. Directionality?*

We corrected the respective sentence which now reads: "For instance, a doubling of the present-day reference value for methane of $1.8\,\mu\text{mol}\,\text{mol}^{-1}$ resulted in a top-of-atmosphere stratospheric adjusted RF of $0.23\,\text{W}\,\text{m}^{-2}$ (Winterstein et al., 2019; Stecher et al., 2021), while studies of Myhre et al. (1998) and Etminan et al. (2016) suggest $0.53\,\text{W}\,\text{m}^{-2}$ and $0.62\,\text{W}\,\text{m}^{-2}$, respectively, for doubling of the reference value of $1.7\,\mu\text{mol}\,\text{mol}^{-1}$."

*2) Section 2.4 CLOUDOPT: Can you provide some details on how the cloud fractions are handled. Do you have separate ice and liquid cloud fractions or are they mixed in a single cloud fraction? How is the vertical overlap of cloud fraction handled? (Maybe a reference for this is sufficient.)*

In CLOUDOPT mass extinction coefficients for ice and liquid clouds are used to calculate the radiative properties (see lines 203-210 in the discussion paper). The cloud fraction, however, is not split into liquid and ice clouds (see the nml in the supplement of Dietmüller et al., 2016). With respect to the cloud overlap we added the following paragraph at the end of the CLOUDOPT section: "In CLOUDOPT and in the radiation schemes the (default) cloud overlap is assumed to be maximum-random overlap (Roeckner et al., 2003; Dietmüller et al., 2016; Giorgetta et al., 2018). In the case of PSrad the overlap assumption is treated based on the Monte Carlo Independent Column Approximation (McICA) technique (see Giorgetta et al., 2018, for details and further references)."

*3) Section 2.5 ALBEDO, line 225: Please define what you mean by "blue-sky", "black-sky" and "white-sky" albedos. In other models, only the direct (your "black-sky" I think) and diffuse (your "white-sky") albedos are needed as the radiation scheme will solve for the direct and diffuse fluxes separately. Presumably the radiation schemes here don't do this and require a combined "blue-sky" albedo as well?*

You are right, we use the terms white-sky and black-sky albedo which are relevant for the direct beam and isotropic diffuse radiation (Liu et al., 2009). The definitions are given in the papers referenced in L225. We have adapted this paragraph which now reads: "In particular, ALBEDO calculates a blue-sky albedo ($\alpha_{blue}$) from the black-sky ($\alpha_{black}$) and white-sky albedo ($\alpha_{white}$) and the fraction of direct and diffuse radiation fluxes with respect

to the total downwelling shortwave fluxes at the surface ($f_{sw,surf}^{dir}$, $f_{sw,surf}^{dif}$) as $\alpha_{blue} = f_{sw,surf}^{dir}\,\alpha_{black} + f_{sw,surf}^{dif}\,\alpha_{white}$ (see e.g. Liu et al., 2009; Li et al., 2018; Cordero et al., 2021, and references therein for details on the different albedos and how to typically derive the blue-sky albedo). Here, the black-sky albedo relates to the albedo associated with the collimated beam, whereas the white-sky albedo corresponds to the albedo associated with isotropic diffuse radiation (Liu et al., 2009)."

Both radiation schemes separate between direct and diffuse flux as noted by Roeckner et al. (2003); Giorgetta et al. (2013). In the latter reference actually RRTMG is described, however PSrad was built based on RRTMG (Pincus and Stevens, 2013). In fact as explained in the text (and as you note in your comment below), the direct and diffuse fluxes are used to calculate the blue-sky albedo (see e.g. line 274 in the discussion paper). With some additional changes it would also be possible for us to pass the direct and diffuse albedos to the radiation schemes. This was, however, not considered in our current simulations but is a potential point of further investigation.

*4) Section 2.5 ALBEDO: There is no mention of the spectral dependence of albedo. How is this handled by these schemes?*

We do not apply any spectral dependent albedo neither in E5rad nor in PSrad. However, e.g. for PSrad we know that both direct and diffuse albedo can be separated into near-infrared and a UV-visible part. As stated before this might be an additional point for further investigation.

*5) Section 2.5 Solar zenith angle dependent albedo, line 277: it would be good to explain at this point that you mean the fraction of diffuse and direct flux will be needed from a previous timestep call of the radiation scheme. What happens at model start-up when there is no previous call?*

We added the respective information and also included the information that in the first model time step the partitioning of 0.9 (direct, black-sky) and 0.1 (diffuse, white-sky) albedo is used to calculate the blue-sky albedo. "To be able to use this new feature, either the radiation scheme has to provide (the fraction of) the direct and diffuse SW radiation fluxes from the previous model time step (for the first model time step the partitioning is automatically set to 0.9 and 0.1, respectively) or ..."

237

*6) Section 2.6 (1): This appears to be an arbitrary functionality to add that could only degrade the physical accuracy of the results. Using the middle of the interval would appear to be the best of the options available. However, none of these options appear to consider what happens when the sun rises or sets during the radiation timestep. I believe the best approach (particularly for solar zenith angle) is to calculate the orbital parameters as a mean over the period of the timestep for which the sun is above the horizon. Was this considered?*

We agree that this functionality seems odd without additional explanation: We included the new offset because we think it is the most reasonable. We kept the old implementation for backward compatibility. Further we added the option to select the offset freely for offline radiation calculations.

We adjusted the respective part: "Now, the offset type can be selected via a new namelist switch. Apart from the previous choice $\Delta t_{orb,opt0}$, which we kept to ensure backward compatibility, the orbital parameters now can be chosen to be calculated for the middle of the interval of time steps associated with the current radiation call $(t_{r,i-1}, t_{r,i-1} + \Delta t_m,..., t_{r,i} - \Delta t_m$, leading to $\Delta t_{orb,opt1} = \frac{1}{2}((t_{r,i} - \Delta t_m) - t_{r,i-1})$, Fig. 2b), or the offset can be set to an arbitrary constant $(\Delta t_{orb,con} \leq \Delta t_r)$. The latter option was introduced for offline radiation calculations."

Regarding the problem of the rising or setting sun: For the radiation calculation the SZA is corrected such that its cosine cannot fall below a certain threshold (see equation 11.23 of Roeckner et al., 2003). Hence, the radiation is calculated globally with at least a certain minimum solar irradiation and later on corrected with the actual SZA (see equation 11.4 of Roeckner et al., 2003). We have incorporated this information in the respective section: "The results from this radiation call (with the adjusted orbital parameters) are later on corrected with the solar irradiation associated with the orbital parameters of the actual model time step for the calculation of the actual SW fluxes and heating rates (see Roeckner et al., 2003). We note that the adjusted SZA contains a modification which ensures that fluxes are non-zero globally to avoid problems in the grid boxes in which the sun rises or sets during the time steps associated with the radiation time step (see Roeckner et al., 2003, ; also their Eq. 11.23).
"

*7) Section 2.6 (2), lines 293-296: Not much point mentioning this adjustment unless you are going to explain how it was adjusted.*

We removed the respective paragraph.

*8) Section 3.1, line 340: It would be useful to give an approximate horizontal resolution in km for T42.*

We rephrased the sentence: "The simulations were conducted with T42 spectral truncation (corresponding to about $2.8° \times 2.8°$, i.e. roughly $300 \, \text{km} \times 300 \, \text{km}$ at the equator) and 90 vertical levels extending up to roughly $80 \, \text{km}$ (see the T42L90MA setup e.g. mentioned by Jöckel et al., 2016)." We also added information on the time step length and the frequency of radiation calls, which we missed to give in the discussion version of the paper.

*9) Section 3.1, line 357: "purely dynamic": I'm not sure what this means (in our usage, this would mean all the physics parametrisations are turned off, which is not the case here).*

We thank the reviewer for pointing out this sloppy use of "dynamic". We have rephrased all such statements referring to the setups at hand as being of "GCM-type".

*10) Section 3.2, paragraph at lines 433-444: I notice you specifically target clear-sky SW with albedo adjustments, but there is nothing to specifically target clear-sky LW. Is surface emissivity fixed for these schemes? Is there anything else that could be used to target this?*

We thank the reviewer for pointing out this possibility. In principle it seems that the radiation schemes could deal with spectrally dependent and regionally varying surface emissivities. However, this is not a feature that is available. We would need to implement additional infrastructure to provide such an emissivity field to the radiation schemes and we would need to acquire the respective data beforehand. Hence, in our simulation we used our standard globally fixed surface emissivity of 0.996 as described by Roeckner et al. (2003). Apart from the surface emissivity we do not see any justifiable "tuning" parameter for clear-sky LW fluxes.

*11) Section 4, line 550: Please explain how the stratospheric adjustment is done.*

The stratospheric adjustment is calculated as described by Stuber et al. (2001) as stated in line 55 of the discussion paper. We have added this information also to the sentence in Section 4: "Table 5 lists the respective perturbations that are calculated in the multiple calls of the radiation scheme. In total, 22 additional (diagnostic) calls for calculating instantaneous RF (calls 02 to 23) and 11 additional calls for calculating stratospheric adjusted RF (calls 24 through 34, where stratospheric adjustment is calculated as described by Stuber et al., 2001), were conducted."

*12) Section 4, line 619-620: "we assumed the 2014 values used by Pincus et al are similar to Meinshausn": I believe the values used by Pincus et al. are essentially those publicly available for RFMIP, so this assumption could be properly checked.*

Pincus et al. (2020) mention that they use 2014 values from "NOAA greenhouse gas inventories". From this information we could not find the reference and the corresponding values and hence we assumed that they are close to the 2014 values presented by Meinshausen et al. (2017).

*13) Section 4, line 628: the N2O RF presented by Pincus should be stated for comparison (even better, all the values from Pincus should be added to table 7).*

We have added the respective value in the text and for the $CO_2$-folding experiments we added a new figure.

*Typos etc.:*

*1) line 11: "of sixth generation of the the" $\rightarrow$ "of the sixth generation of the"*

Done.

*2) line 55: "radiative RFs" $\rightarrow$ "RFs"*

Done.

*3) line 86: "old radiation" → "old radiation scheme"*

Done.

*4) line 351: table 2 is referenced before table 1*

Thank you for spotting this inconsistency. We rearranged the tables.

*5) line 430: "adjust parameters target-oriented" → "adjust parameters in a target-oriented manner"*

Done.

*6) line 679: "much increased (decreased) to the radiative forcings" → "much increased (decreased) with respect to the radiative forcings"*

We adjusted the sentence to "...much increased (decreased) in comparison to the radiative forcings...".

**Reply to comments from Referee #2**

**(https://doi.org/10.5194/egusphere-2023-2140-RC2)**

Below we will address all comments of referee #2 and will state corresponding changes in the manuscript. Again, we would like to thank referee #2 for taking the time to review our manuscript.

*This manuscript describes major updates to the radiation schemes within the Modular Earth Submodel System (MESSy), which is an infrastructure designed to link different submodels into the same framework to more seamlessly perform simulations with different model components. Specifically, this work covers the implementation of the PSrad radiation scheme into MESSy, as well as updates to related submodels for calculating cloud optical properties (CLOUDOPT) and aerosol optical properties (AEROPT), as well as implementation into MESSy of a new albedo scheme (ALBEDO). The authors find that implementation of these schemes leads to reduced biases in temperature and humidity of a handful of key climate processes and improvement in radiative forcing variables for greenhouse gases relative to reference values. I find it particularly valuable that the implementation allows for easier calculation of radiative forcing through online double calls. These calculations are important but not routinely performed at most modeling centers. This manuscript is well written and will be of interest to GMD readers, especially as many modeling centers work towards updating their radiation schemes and, more generally, work towards stronger unification of submodels. I recommend some minor revisions detailed below.*

We thank the reviewer for this rating of our manuscript. We will address all minor revisions suggested below.

*General: I think readers would appreciate some information about computational performance when implementing the new radiative transfer scheme with more spectral bands. Was there a noticible increase in compute time with the new code and, if so, what steps did the developers take in an attempt to improve speeds?*

We thank the reviewer for pointing out that this information was lacking in the manuscript. The computational time for the GCM-type simulation increased by 70%, however this increase is due to the combined effect of the "old" vs the " new" setups, i.e. it includes also possible increases in computational time from the other updated submodels: AEROPT, CLOUDOPT, ALBEDO. For simulations with full chemistry, which we typically aim at, this increase will not play a major role due to the large computational demand of the chemistry solver. We have added a corresponding paragraph at the end of Section 3.1.: "Without additional diagnostic radiation calls for RF calculations as presented in Section 4, for a simulation performed on a single node[1] the computational time required for a radiation time step is around 70% higher for the PSrad setups than for the E5rad setups. If the full radiation calls are only performed every third time step (as in the simulation setups described above), this leads to an increase in the computational time of roughly 40%. This increase in computational time cannot be solely attributed to the core radiative transfer routines in RAD but is also affected by possible changes in computational time in the connected submodels AEROPT, CLOUDOPT and ALBEDO. To put this increase into perspective, we note that EMAC is commonly used in setups with comprehensive interactive chemistry (e.g. as chemistry-climate model). Due to the large computational demand of the chemistry solver the increase in computational time due to the radiation scheme will only be a fraction of the increase we report here for a GCM-type setup."

Footnote:"[1] 32 task on an AMD Epyc 7601 node with 32 cores"

*Line 206-207: It may be a bit surprising to some, me included, that the developers decided to add a secondary LW ice mass extinction option that comes from a model that is now a few generations old (ECHAM4). What there a particularly reason to bring back this scheme? Some context here would be interesting.*

We are sorry for the impression that we newly implemented this feature. It has been an option of the MESSy submodel CLOUDOPT before and we simply kept it for backward compatibility reasons. We slightly rephrased the sentence by changing "also allows" to "still allows" to make clear that this option was not introduced during our development but simply preserved.

*Line 245-258: What is the role of this observational-based albedo climatology when the scheme is used to simulate climates beyond the present-day? Is the climatology used as a scaling factor to preserve seasonality? Is it only implemented for certain types of simulations?*

Indeed the observational based albedo was not changed for our pi and pd simulations and it is not routinely implemented to use it to modify (transient) albedos associated with different climate states (e.g. concerning land-use change). However, we note that it is only the background albedo and is modified e.g. by the snow cover (see lines 268-275 in the discussion version). Before our implementations we have used an old background albedo from ECHAM5, which did not feature a seasonal cycle. Further, if a certain transient albedo associated with a specific scenario would be available, it could be easily applied with the new submodel ALBEDO (see lines 240-243 in the discussion version).

*Section 2.6-1: Some motivation for providing additional flexibility in the orbital offset would be helpful. The previous version, where the offset would always falls in the middle between radiation calls, seems like the most reasonable approach for any case. Are there cases where another option is better? Some context would be helpful here.*

We agree that some more motivation is needed. We have introduced the new option (middle between time steps associated with the respective radiation call), which we think is most suitable. We understand that referee #1 agrees on that. The previous option (middle between radiation steps) was preserved for backward compatibility. The freely adjustable option is important for offline radiation calculation purposes. In response to this comment and the comment by reviewer #1 (see minor comment 6) we adjusted the section as follows: "Now, the offset type can be selected via a new namelist switch. Apart from the previous choice $\Delta t_{orb,opt0}$, which we kept to ensure backward compatibility, the orbital parameters now can be chosen to be calculated for the middle of the interval of time steps associated with the current radiation call ($t_{r,i-1}$, $t_{r,i-1} + \Delta t_m$,..., $t_{r,i} - \Delta t_m$, leading to $\Delta t_{orb,opt1} = \frac{1}{2}((t_{r,i} - \Delta t_m) - t_{r,i-1})$, Fig. 2b), or the offset can be set to an arbitrary constant ($\Delta t_{orb,con} \leq \Delta t_r$). The latter option was introduced for offline radiation calculations."

*Line 355: It is clear that the sets of simulations performed in this section have different radiation schemes (PSrad vs E5rad) but what about the modifications to the other relevant submodels discussed? I suspect the simulations using of PSrad also include all of the updates discussed for CLOUDOPT, AEROPT, ALBEDO and the orbital offset. If so, this should be noted in the text or better*

*incorporated into the experiment names for clarity.*

We agree that the previous formulation at the beginning of Section 3.1 was not clear about this. Hence we adapted it: "We performed four simulations for the evaluation presented here. Namely, two simulations (pre-industrial and present-day denoted with pi and pd, respectively) for each of the two radiation schemes (the old ECHAM5 radiation scheme with the v2 in the SW, denoted here with E5rad, and the newly implemented PSrad scheme). These simulations will be addressed here as EMAC-E5rad-pi, EMAC-E5rad-pd, EMAC-PSrad-pi and EMAC-PSrad-pd, respectively. The simulation setups do not differ only in the radiation scheme but also according to the respective radiation scheme the typical old and new setups of AEROPT, CLOUDOPT and ALBEDO (as described before) have been chosen as indicated in Table 1. In all simulations the new choice for the orbital offset parameter ($\Delta t_{orb}$) was employed."
Information on the setup is partly also contained in Table 1 (previous Table 2). We now explicitly refer to this table and also adapted it such that the setups can be followed more easily.

*General Section 3: The biases are presented clearly, and the authors focus on important ones, but I was hoping for some attempt to explain the causes of the bias, and particularly for situations where the e5rad and Psrad-driven simulation biases differ. Establishing causation is difficult in many cases, but some general discussion or potential explanations from the authors would be useful here. Is the warm stratosphere bias from the PSrad simulation (compared to the cold bias from the e5rads) related to the new handling of the orbital parameter offset, for instance?*

We are glad that our comparison is presented clearly. We can also understand the wish to establish causality. However, from the simulations at hand this is difficult to do. We would need to setup additional experiments to disentangle the different effects due to changes of the albedo or the different tropospheric aerosol etc. We compare our results to the changes from ECHAM5 to ECHAM6 presented by Stevens et al. (2013), which are related to changes in the radiation scheme. But also in this study not only the radiation scheme was changed but at several instances the model was updated. However, we can rule out that the orbital parameter offset is causing this effect because the new choice for this parameter was used in all simulation that we present in the paper. Now we

516 mention this fact also in the text (see our reply to your previous question)

517

518 *Also relevant to Figure 5: ERA5 has a known cold bias in stratospheric*
519 *temperature from 2000 to 2006, The reanalysis was rerun for this period in a*
520 *product called ERA5.1. I am unfamiliar with how large this bias was, but it*
521 *would be interesting to see if the EMAC-PSrad bias is reduced for years outside*
522 *of this range, or if ERA5.1 is used instead. Presumably the Figure 7 humidity*
523 *bias is impacted too. Details here:* `https: // confluence. ecmwf. int/ pages/`
524 `viewpage. action? pageId= 181130838`

525

526 We thank the reviewer for pointing this out. In the discussion version we
527 did not add a note regarding ERA5.1 to avoid any confusion. In response to
528 this comment, we decided to add the following paragraph after the comparison
529 of specific humidity from ERA5 and our simulations: "Due to a setup incon-
530 sistency ERA5 has a cold bias in the stratosphere for the period 2000 to 2006,
531 which also affects stratospheric water vapour (Simmons et al., 2020). This issue
532 has been addressed in a new set of analyses called ERA5.1 covering this period
533 (Simmons et al., 2020). We note however, that the differences between ERA5.1
534 and ERA5 regarding temperatures and water vapour as analysed by Simmons
535 et al. (2020) are relatively small compared to the differences we see between
536 ERA5 and our model simulations. Hence we simply applied the ERA5 data
537 as the main conclusions regarding the model reanalyses differences will remain
538 unchanged."

539

540 *Line 599-606: Is the reduction in methane RF from IRF for PSrad sig-*
541 *nificant? A 0.01 W/m2 reduction from IRF seems quite small and may just be*
542 *noise, especially when the reduction does not appear to be present for the pi sim-*
543 *ulation. I mention this because although stratospheric adjustments related to SW*
544 *absorption may be playing a role in a reduction, the Smith et al figure points to*
545 *cloud adjustments playing in even larger role, an effect not being captured in this*
546 *work. And recently, Allen et al. 2023 looked into the cooling from SW absorption*
547 *of methane explicitly, finding much of it is driven by cloud adjustments, rather*
548 *than a stratospheric adjustment: https://www.nature.com/articles/s41561-023-*
549 *01144-z*

550

551 We thank the reviewer for this comment. We have adjusted the respective
552 section as follows: "Another aspect to note about the methane RFs is that

with E5rad the stratospheric temperature adjustment acts to increase the RF in comparison to the instantaneous RF, whereas for PSrad the differences between instantaneous and stratospheric adjusted RF are smaller and the sign depends on the background state. PSrad includes SW absorption of methane in two bands in the near-infrared (3.08 - 3.85 $\mu$m and 2.15 - 2.50 $\mu$m; cf. the RRTM bands described in the ECHAM6 documentation Giorgetta et al., 2013). The SW absorption acts to counteract the stratospheric cooling induced by the LW radiation (Byrom and Shine, 2022, their Fig. 2). Hence, the adjustment difference we find between PSrad and E5rad is in part consistent with the results from Smith et al. (2018, their Fig. S6). They point out that for the same experiments as analysed by Richardson et al. (2019), the rapid radiative adjustment induced by the stratospheric temperature adjustment is negative in models with the explicit treatment of methane SW absorption in the radiation scheme, and positive in models without. However, in the latter case the increase reported by Smith et al. (2018) is more pronounced as there is a substantial additional contribution from cloud radiative adjustments that are not covered by our technique."

*Line 638-639: Yes, the Pincus pd background likely has a warmer surface thus CO2 forcing is stronger, but it also likely has a cooler stratosphere, which is arguably more impactful on CO2 forcing as highlighted by Jeevangee et al. 2021 and He et al. 2023. Related, this may explain why the CO2 forcing from the PSrad simulation is smaller than the E5rad simulations. PSrad produces a warmer stratosphere and thus the CO2 forcing is smaller.*

*Jeevangee et al. 2021: https://doi.org/10.1175/JCLI-D-19-0756.1*

*He et al. 2023: https://www.science.org/doi/10.1126/science.abq6872*

We thank the reviewer for this comment. Regarding the impact of the stratosphere, we have adjusted the respective section: "(ii) In the climatological pd background, the tropospheric temperatures are likely higher and the stratospheric temperatures lower than for our pi background. Here, we reason that both changes will likely lead to an increased RF as diagnosed from $CO_2$-folding experiments, with the stratospheric component potentially making the larger contribution (He et al., 2023)"

Regarding the second part of the comment: We thank the reviewer for pointing this out. Indeed it seems that this could contribute to the differences. This can

be inferred from comparing our Table 6 with Table 8, where the latter shows all-sky instantaneous RFs when the radiation scheme is switched compared to the radiation scheme that drives the model simulation. Hence, we added a new paragraph after the introduction of Table 8: "Related to the dependence of RFs for $CO_2$ perturbations on the background, we have previously detected a larger $CO_2$ sensitivity in the E5rad compared to the PSrad simulations. As discussed above for the dependence of the instantaneous $CO_2$ RFs on the pi and pd background, we point out that a warmer stratosphere in the PSrad compared to the E5rad simulations might be contributing to the lower RF values diagnosed from PSrad compared to E5rad. In line with this argument, instantaneous all-sky $CO_2$ RFs increase (decrease) for E5rad (PSrad) when the background is provided by the switched radiation scheme PSrad (E5rad) as can be seen from the comparison of Tables 6 and 8."

653  Preliminary assessment of 20-m surface albedo retrievals from sentinel-2a
654  surface reflectance and modis/viirs surface anisotropy measures. *Remote*
655  *Sensing of Environment*, 217:352–365, 2018. ISSN 0034-4257. doi: 10.
656  1016/j.rse.2018.08.025. URL `https://www.sciencedirect.com/science/`
657  `article/pii/S0034425718304024`.

658  J. Liu, C. Schaaf, A. Strahler, Z. Jiao, Y. Shuai, Q. Zhang, M. Roman, J. A.
659  Augustine, and E. G. Dutton. Validation of moderate resolution imaging spec-
660  troradiometer (modis) albedo retrieval algorithm: Dependence of albedo on
661  solar zenith angle. *Journal of Geophysical Research: Atmospheres*, 114(D1),
662  2009. doi: 10.1029/2008JD009969. URL `https://agupubs.onlinelibrary.`
663  `wiley.com/doi/abs/10.1029/2008JD009969`.

664  M. Meinshausen, E. Vogel, A. Nauels, K. Lorbacher, N. Meinshausen, D. M.
665  Etheridge, P. J. Fraser, S. A. Montzka, P. J. Rayner, C. M. Trudinger, P. B.
666  Krummel, U. Beyerle, J. G. Canadell, J. S. Daniel, I. G. Enting, R. M. Law,
667  C. R. Lunder, S. O'Doherty, R. G. Prinn, S. Reimann, M. Rubino, G. J. M.
668  Velders, M. K. Vollmer, R. H. J. Wang, and R. Weiss. Historical greenhouse
669  gas concentrations for climate modelling (cmip6). *Geoscientific Model De-*
670  *velopment*, 10(5):2057–2116, 2017. doi: 10.5194/gmd-10-2057-2017. URL
671  `https://gmd.copernicus.org/articles/10/2057/2017/`.

G. Myhre, E. J. Highwood, K. P. Shine, and F. Stordal. New estimates of radiative forcing due to well mixed greenhouse gases. *Geophysical Research Letters*, 25(14):2715–2718, July 1998. doi: 10.1029/98GL01908.

R. Pincus and B. Stevens. Paths to accuracy for radiation parameterizations in atmospheric models. *Journal of Advances in Modeling Earth Systems*, 5(2):225–233, 2013. doi: 10.1002/jame.20027. URL https://agupubs. onlinelibrary.wiley.com/doi/abs/10.1002/jame.20027.

R. Pincus, S. A. Buehler, M. Brath, C. Crevoisier, O. Jamil, K. Franklin Evans, J. Manners, R. L. Menzel, E. J. Mlawer, D. Paynter, R. L. Pernak, and Y. Tellier. Benchmark calculations of radiative forcing by greenhouse gases. *Journal of Geophysical Research: Atmospheres*, 125 (23):e2020JD033483, 2020. doi: 10.1029/2020JD033483. URL https: //agupubs.onlinelibrary.wiley.com/doi/abs/10.1029/2020JD033483. e2020JD033483 10.1029/2020JD033483.

T. B. Richardson, P. M. Forster, C. J. Smith, A. C. Maycock, T. Wood, T. Andrews, O. Boucher, G. Faluvegi, D. Fläschner, Hodnebrog, M. Kasoar, A. Kirkevåg, J.-F. Lamarque, J. Mülmenstädt, G. Myhre, D. Olivié, R. W. Portmann, B. H. Samset, D. Shawki, D. Shindell, P. Stier, T. Takemura, A. Voulgarakis, and D. Watson-Parris. Efficacy of climate forcings in pdrmip models. *Journal of Geophysical Research: Atmospheres*, 124(23): 12824–12844, 2019. doi: 10.1029/2019JD030581. URL https://agupubs. onlinelibrary.wiley.com/doi/abs/10.1029/2019JD030581.

E. Roeckner, G. Bäumel, L. Bonaventura, R. Brokopf, M. Esch, M. Giorgetta, S. Hagemann, I. Kirchner, L. Kornblueh, E. Manzini, A. Rhodin, U. Schlese, U. Schulzweida, and A. Tompkins. The atmospheric general circulation model ECHAM5, PART I, Model description. *Report / Max-Planck-Institut für Meteorologie*, 349, 2003. doi: 10.17617/2.995269.

A. Simmons, C. Soci, J. Nicolas, B. Bell, P. Berrisford, R. Dragani, J. Flemming, L. Haimberger, S. Healy, H. Hersbach, A. Horányi, A. Inness, J. Munoz-Sabater, R. Radu, and D. Schepers. Global stratospheric temperature bias and other stratospheric aspects of era5 and era5.1, 01/2020 2020. URL https: //www.ecmwf.int/node/19362.

C. J. Smith, R. J. Kramer, G. Myhre, P. M. Forster, B. J. Soden, T. Andrews, O. Boucher, G. Faluvegi, D. Fläschner, Hodnebrog, M. Kasoar, V. Kharin,

A. Kirkevåg, J.-F. Lamarque, J. Mülmenstädt, D. Olivié, T. Richardson, B. H. Samset, D. Shindell, P. Stier, T. Takemura, A. Voulgarakis, and D. Watson-Parris. Understanding rapid adjustments to diverse forcing agents. *Geophysical Research Letters*, 45(21):12,023–12,031, 2018. doi: 10.1029/2018GL079826. URL https://agupubs.onlinelibrary.wiley.com/doi/abs/10.1029/2018GL079826.

L. Stecher, F. Winterstein, M. Dameris, P. Jöckel, M. Ponater, and M. Kunze. Slow feedbacks resulting from strongly enhanced atmospheric methane mixing ratios in a chemistry–climate model with mixed-layer ocean. *Atmospheric Chemistry and Physics*, 21(2):731–754, 2021. doi: 10.5194/acp-21-731-2021. URL https://acp.copernicus.org/articles/21/731/2021/.

B. Stevens, M. Giorgetta, M. Esch, T. Mauritsen, T. Crueger, S. Rast, M. Salzmann, H. Schmidt, J. Bader, K. Block, R. Brokopf, I. Fast, S. Kinne, L. Kornblueh, U. Lohmann, R. Pincus, T. Reichler, and E. Roeckner. Atmospheric component of the mpi-m earth system model: Echam6. *Journal of Advances in Modeling Earth Systems*, 5(2):146–172, 2013. doi: 10.1002/jame.20015. URL https://agupubs.onlinelibrary.wiley.com/doi/abs/10.1002/jame.20015.

N. Stuber, R. Sausen, and M. Ponater. Stratosphere adjusted radiative forcing calculations in a comprehensive climate model. *Theoretical and Applied Climatology*, 68:125–135, 2001. doi: 10.1007/s007040170041. URL https://doi.org/10.1007/s007040170041.

[revised manuscript text omitted]

---

## Author Response (AR2)

**Author Comment to the revised version of the manuscript egusphere-2023-2140, (https://doi.org/10.5194/egusphere-2023-2140, in review, 2023): "Updating the radiation infrastructure in MESSy (based on MESSy version 2.55)"**

by M. Nützel et al.

March 21, 2024

**To the editor**

Dear editor, thank you for accepting our manuscript subject to minor revisions based on the reviewer's comment. We would like to note that there was likely a misunderstanding regarding the orbital offset. We will try to clarify this by adjusting the manuscript and by providing a detailed response (blue) to the reviewer's comment (*black italics*) here. We will also provide a manuscript which will highlight the changes in comparison to the revised version. We thank the referee for taking the time to review the revised version of our paper and the editor for handling the manuscript.

**Reply to reviewer**

We thank the reviewer for agreeing to check the revised version of our manuscript. The reviewer raised one concern regarding the orbital offset used in the full calculation of the radiation. We think that there is still a misunderstanding which

we hope to address satisfactorily by replying to this comment and by adapting the text of our manuscript. The issue raised is as follows:

*However, one minor, but important comment (from both reviewers) has been misunderstood. Section 2.6 (1): the original offset (opt0, old default) correctly uses the middle of the time interval for the calculation of the solar zenith angle. I believe the adoption of the new default (opt1) will introduce an error into the calculation. The radiation calculation needs to represent the movement of the sun over the whole period from the beginning to the end of the radiation timestep, not a discrete point in the interval. Therefore, opt0 (old default) is a better choice if you are going to select a representative time. (Even better would be to actually calculate the mean value of the solar zenith angle over the interval, but I appreciate this is beyond the scope of this paper.)*

We are sorry that we misinterpreted the initial review comment regarding this issue and thank the reviewer for clarifying the disagreement, which we try to resolve now.

Maybe in our last reply we also did not make clear that the full radiation calculation is later on corrected with the exact orbital parameters of the model time steps associated with the full radiation call. To make this easier to follow we give the following example: Let's assume that the full radiation is performed every 30 minutes while the model time step is 10 minutes. At 10:00 a full radiation calculation is performed and the next one follows at 10:30. Previously, the offset was 15 minutes (30 minutes/2), hence the orbital parameters at 10:15 were used for the full radiation calculation. The results were then corrected using the orbital parameters at 10:00, 10:10 and 10:20 to provide the fluxes and heating rates for these time steps. Which we think is inconsistent as for the time step at 10:30 the results from the next full radiation with orbital parameters set to 10:45 were used. There are two ways of solving this inconsistency: (i) shifting the orbital parameters which are used to correct the full radiation calculation by half a model time step, which would lead to orbital parameters representative of 10:05, 10:15 and 10:25 (maybe this is what you had in mind) or (ii) shifting the initial orbital offset for the full radiation calculation to 10:10 which is (on average) closer to the orbital parameters at 10:00, 10:10 and 10:20 which are used to correct the fluxes and heating rates. I.e. the aim is to find the offset that produces the least error when the corrections with the orbital parameters

at the model time steps associated with the radiation call are used. This does not aim to provide the best offset representative of the orbital parameters for the time span from one full radiation call to the next.

We agree that overall option (i) could be even better. However, option (ii) is still an improvement regarding consistency and the shift of the orbital offset for the correction was never considered before. We also note that the difference in using the old and new default is rather small for the currently applied time step lengths and radiation call frequencies.

We have adjusted the respective parts in the manuscript to better motivate our choice and hope that this is also satisfactory for the reviewer.